# Biologically-plausible backpropagation through arbitrary timespans via local neuromodulators

Yuhan Helena Liu[1,2,3,*], Stephen Smith[2,4], Stefan Mihalas[1,2,3], Eric Shea-Brown[1,2,3], and Uygar Sümbül[2,*]

[1]Department of Applied Mathematics, University of Washington, Seattle, WA, USA
[2]Allen Institute for Brain Science, 615 Westlake Ave N, Seattle WA, USA
[3]Computational Neuroscience Center, University of Washington, Seattle, WA, USA
[4]Department of Molecular and Cellular Physiology, Stanford University, Stanford CA, USA
[*]Correspondence: hyliu24@uw.edu, uygars@alleninstitute.org

## Abstract

The spectacular successes of recurrent neural network models where key parameters are adjusted via backpropagation-based gradient descent have inspired much thought as to how biological neuronal networks might solve the corresponding synaptic credit assignment problem [1–3]. There is so far little agreement, however, as to how biological networks could implement the necessary backpropagation through time, given widely recognized constraints of biological synaptic network signaling architectures. Here, we propose that extra-synaptic diffusion of local neuromodulators such as neuropeptides may afford an effective mode of backpropagation lying within the bounds of biological plausibility. Going beyond existing temporal truncation-based gradient approximations [4–6], our approximate gradient-based update rule, ModProp, propagates credit information through arbitrary time steps. ModProp suggests that modulatory signals can act on receiving cells by convolving their eligibility traces via causal, time-invariant and synapse-type-specific filter taps. Our mathematical analysis of ModProp learning, together with simulation results on benchmark temporal tasks, demonstrate the advantage of ModProp over existing biologically-plausible temporal credit assignment rules. These results suggest a potential neuronal mechanism for signaling credit information related to recurrent interactions over a longer time horizon. Finally, we derive an in-silico implementation of ModProp that could serve as a low-complexity and causal alternative to backpropagation through time.

## 1 Introduction

Recurrent connectivity is a hallmark of neuronal circuits. While this feature enables rich and flexible computation, mechanisms enabling efficient task learning in large circuits remain a central problem in neuroscience and artificial intelligence research. Fundamentally, the problem stems from the fact that potentially all history of all neurons, including synaptically far away ones, can affect neuronal activity and contribute to task output. Motivated by the success of gradient descent learning, several biological learning models approximate the exact gradient in recurrent neural networks using known biological processes and gain insights into computational principles of how the brain might learn [4–6].

By ignoring dependencies beyond a few recurrent steps — thus severely truncating the gradient computational graph — these existing models succeed in representing their approximate gradient-based update rule as a combination of terms that resemble known synaptic physiological processes: "eligibility trace", which maintains a fading memory of coincidental activation of presynaptic and

postsynaptic neurons [7–11], combined with a third "modulatory" factor — top-down learning or "reward" signals [12–15], for which dopamine is a prominent candidate [16]. Despite the impressive performance of such approximations on a variety of tasks, truncating relevant credit information results in a significant performance gap compared to algorithms using exact gradient information; backpropagation-through-time (BPTT) and real time recurrent learning (RTRL) [4–6, 17].

How might neural circuits account for long-term recurrent interactions to assign credit (or blame) to neuronal firing that happened arbitrary steps before the presentation of reward? Given the rich repertoire of dynamical and signaling elements in the brain, one avenue could be to examine biological processes that have been underexplored in existing models. Dopamine — whose cellular actions are exerted by activation of G protein-coupled receptors (GPCRs), which can greatly impact STDP — is not the only neuromodulator involved in learning [18–21]. More recently, transcriptomic studies have uncovered strong evidence for many other neuromodulatory pathways throughout the brain that also act via the activation of GPCRs, leading to similar downstream actions as dopamine [22, 23]. This suggests that, similar to dopamine, they could also play a role in shaping synaptic credit assignment. A conspicuous family within these pathways is neuropeptide signaling because peptidergic genes are densely and abundantly expressed in the forebrains of divergent species, including the human, in a cell-type-specific fashion [24], suggesting widespread interaction between synaptic and peptidergic modulatory networks for synaptic credit assignment [6, 25]. Moreover, intra-cortical expression allows neuropeptides to potentially carry information **local** to the cortical network, cell type specificity enables sculpted signals for different recipient cells, and their diffusive nature could enable communication between neurons that are not synaptically connected. Finally, peptidergic signals have timescales much longer than the time scales for axonal propagation of action potentials or synaptic delays [26]. Taken together, these properties make cell-type-specific local neuromodulation seem promising for propagating credit signal over multiple recurrent steps. Developing explicit computational principles of how these local modulatory elements could propagate credit signal over arbitrary recurrent steps could advance our understanding of biological learning and may inspire more efficient low-complexity bio-inspired learning rules.

Motivated by the question above as well as shortcomings of gradient approximations based on severe temporal truncations, we investigate how biological credit signals could be propagated through arbitrary recurrent steps via widespread cell-type-specific neuropeptidergic signaling [23, 25]. While Ref. [6] recently introduced a framework exploiting properties of neuropeptidergic signaling for temporal credit assignment, similar to [4] and [5], their approach performs severe temporal truncation of the error gradient and does not consider credit propagation beyond disynaptic connections. Our main contributions are summarized as follows:

- We derive a theory that provides mechanism and intuition for the effectiveness of synapse-type-specific modulatory backpropagation (through time) weights (Theorem 1).
- We develop a model predicting how modulatory signaling could be the basis for biological backpropagation through **arbitrary** time spans (Figures 1 and 2). Unlike temporal truncation-based approximations [4–6], our model enables each neuron to receive filtered (rather than precise) credit signals regarding its contribution to the outcome via neuromodulation.
- We demonstrate the effectiveness of modulatory signaling via synapse-type-specific, rather than synapse-specific, modulatory weights on learning tasks that involve temporal credit assignment (Figure 4). In particular, we demonstrate an **online** learning setting where weights are updated causally and in real-time (Figure 5).
- We also derive a low-complexity in silico implementation of our algorithm suitable for **online learning** (Proposition 1).

## 2 Related works

**Neuromodulatory factors in synaptic plasticity:** One of the most fundamental learning rules, Hebbian plasticity, attributes lasting changes in synaptic strength and memory formation to correlations of spike timing between particular presynaptic and postsynaptic neurons [27, 28]. However, multiple experimental and theoretical investigations now indicate that the Hebbian rule alone is insufficient. First, there have been numerous suggestions that some persistent "eligibility trace" must exist to bridge the temporal gap between correlated firings at millisecond timescales and behavioral

timescales lasting seconds [7–10, 29, 30]. Moreover, impacts of correlated spike timing must be augmented by one or more additional modulatory factors to steer weight updates toward desired outcome [1, 7, 8, 15, 16, 31–38]. This is commonly known as learning with three factors. A prominent candidate for such a modulatory factor is dopamine [13]. Dopamine influences receiving neurons via activation of G protein-coupled receptors (GPCRs), which can regulate membrane excitability and key parameters of synaptic plasticity rules.

Besides dopamine, recent transcriptomic evidence has uncovered genes encoding a plethora of other neuromodulatory pathways throughout the brain, including neuropeptide signaling genes that are abundantly expressed in the forebrains of tetrapods, including the human [23, 24]. Like dopamine, their cellular actions are exerted by the activation of G protein-coupled receptors (GPCRs), which can persistently modulate Hebbian synaptic plasticity [15, 39, 40]. This suggests that they too, could play a role in shaping cortical learning and synaptic credit assignment. Moreover, nearly all neurons express one or more neuropeptide signaling gene [23], which suggests a dense interplay between synaptic and peptidergic modulatory networks to shape synaptic credit assignment [25].

**Approximate gradient descent learning:** Standard algorithms for gradient descent learning in recurrent neural networks (RNNs), real time recurrent learning (RTRL) and backpropagation through time (BPTT), are not biologically-plausible [41] and have vast computational and storage demands [42]. However, multiple studies have shown that learning algorithms that approximate the gradient, while mitigating some of the problems of exact gradient computation, can lead to satisfactory learning outcomes [41, 43, 44]. In feedforward networks, plenty of biologically-plausible learning rules have been proposed and demonstrated impressive performance that rival backpropagation on many different tasks [14, 33, 34, 41, 44–49].

For efficient online learning in RNNs, approximations to RTRL have been proposed [4, 17, 50–54]. For instance, a recent influential study [5] conceived how the three-factor learning framework could approximate the gradient. Among the biologically-plausible proposals [4–6], approximations have mainly been based on temporal truncations of the gradient computation graph and because of that, their ability to learn dependencies across arbitrary recurrent steps has been limited. Lastly, non-truncation-based approximations have been proposed outside of bio-plausible research [50, 51]. For further discussions on related algorithms, please refer to Appendix C.

## 3 Results

### 3.1 Learning rule overview

Biological plausibility is a guiding principle in developing our model. Hence, the model choices are based either on established constraints of neurobiology such as the locality of synaptic transmission, causality, and Dale's Law, or on emerging evidence from large-scale datasets, such as the cell-type-specificity of certain neuromodulators [23] or the hierarchical organization of cell types [22]. We consider a discrete-time rate-based RNN similar to the form in [56] with observable states, i.e. firing rates, as $z_t$ at time $t$, and the corresponding internal states as $s_t$. $W$ denotes the recurrent weight matrix with $(pq)^{\text{th}}$ entry $W_{pq}$ representing the synaptic connection strength between presynaptic neuron $q$ and postsynaptic neuron $p$. See Supp. Section A for the neuron model.

**Gradient descent learning in RNNs**: RNNs are typically trained by gradient descent learning on the task error (or negative reward) $E$. However, the two equivalent factorizations of error gradient in RNNs, BPTT and RTRL, both involve nonlocal information that is inaccessible to neural circuits. This is due to recurrent connectivity: a synaptic weight $W_{pq}$ can affect loss $E$ through many other neurons in the other network in addition to its pre- and postsynaptic neurons. To see this more concretely, the following is the exact gradient at time $t$ using RTRL factorization, on which we base our approximation for online learning:

$$\frac{\mathrm{d}\,E}{\mathrm{d}\,W_{pq}}|_t = \sum_j \frac{\partial E}{\partial s_{j,t}} \frac{\mathrm{d}\,s_{j,t}}{\mathrm{d}\,W_{pq}} \tag{1}$$

$$\frac{\mathrm{d}\,s_{j,t}}{\mathrm{d}\,W_{pq}} = \frac{\partial s_{j,t}}{\partial W_{pq}} + \sum_l \frac{\partial s_{j,t}}{\partial s_{l,t-1}} \frac{\mathrm{d}\,s_{l,t-1}}{\mathrm{d}\,W_{pq}} = \frac{\partial s_{j,t}}{\partial W_{pq}} + \frac{\partial s_{j,t}}{\partial s_{j,t-1}} \frac{\mathrm{d}\,s_{j,t-1}}{\mathrm{d}\,W_{pq}} + \underbrace{\sum_{l \neq j} W_{jl} \frac{\partial z_{l,t-1}}{\partial s_{l,t-1}} \frac{\mathrm{d}\,s_{l,t-1}}{\mathrm{d}\,W_{pq}}}_{\text{depends on all weights } W_{jl}}, \tag{2}$$

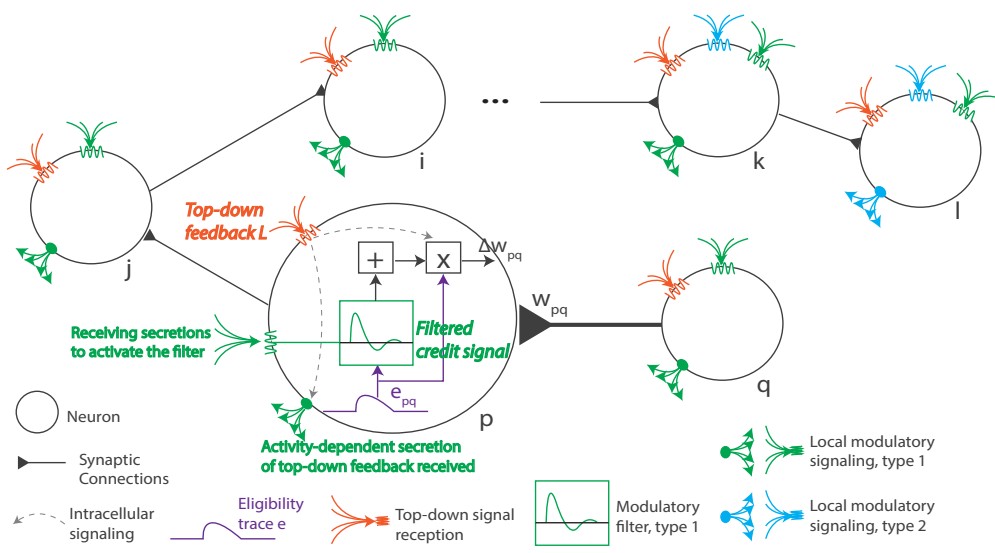

Figure 1: **Biologically-plausible temporal credit assignment via modulatory and synaptic message passing.** In addition to established biological learning ingredients (eligibility traces and a top-down learning signal [8, 55]), synapse-type-specific local modulatory networks may also be involved in weight updates [25]. Our learning rule, ModProp, conceives the action of participating modulators on receiving cells as a convolution of the eligibility trace with causal, time-invariant, cell-type-specific filters. Each circle represents a neuron and the synaptic weight of interest is $W_{pq}$; we illustrate the cellular processes of postsynaptic neuron $p$. Our derivation (Supp. Section A and Theorem 1) predicts that the modulatory signal each neuron receives can represent a filtered credit signal regarding how its past firings (arbitrary steps back) contribute to the task outcome.

where $\partial$ denotes direct dependency and $d$ accounts for all (direct and indirect) dependencies, following the notation in [5]. (See Appendix A.2 for further details on the notation.) While $\frac{\partial E}{\partial s_{j,t}}$, which considers only the direct contribution of the internal state of neuron $j$ at time $t$ to the loss, is easy to compute, the factor $\frac{d\,s_{j,t}}{d\,W_{pq}}$ is a memory trace of all inter-cellular dependencies and requires $O(N^3)$ memory and $O(N^4)$ computations. This makes RTRL expensive to implement for large networks. Moreover, the last factor $\frac{d\,s_{j,t}}{d\,W_{pq}}$ poses a serious problem for biological plausibility: the nonlocal terms in Eq. 2 requires knowledge of all other weights in the network to update the weight $W_{pq}$. Existing biologically-plausible solutions to this problem apply severe truncations: references [4] and [5] completely ignore the third nonlocal term (Figure 2), whereas reference [6] restores terms within one recurrent step through local modulatory signaling but truncates further terms.

**ModProp approximation overview:** Derivation of gradient descent-based weight updates involving neuromodulatory signaling from Eq. 1 suggests two approximations to Eq. S4 and 2 to move beyond severe truncations of the exact gradient while retaining biological plausibility. **Approximation 1** replaces the activation derivative with a constant:

$$\frac{\partial z_{j,t}}{\partial s_{j,t}} \approx \mu \quad \text{(approximation 1)}, \tag{3}$$

where $\mu$ represents the average activity of neurons in a ReLU network. This approximation assumes stationarity in neuron activity and uncorrelatedness of such activity with a small subset of synaptic weights, as explained in derivations leading up to Appendix Eq. S14 and Eq. S21 (Methods in Appendix A). While neuronal activity and synaptic weights are not necessarily uncorrelated, considering that a single neuron may have thousands of synaptic partners, the activity of the neuron or its time derivative is typically weakly correlated to any one synaptic weight. This also indicates that such approximation might work better for large networks with many neurons and synaptic partners for each neuron, as is the case for biological neural networks. We take advantage of this phenomenon in our model and ignore these weak correlations. This approximation enables the filter taps (Eq. 6) to be

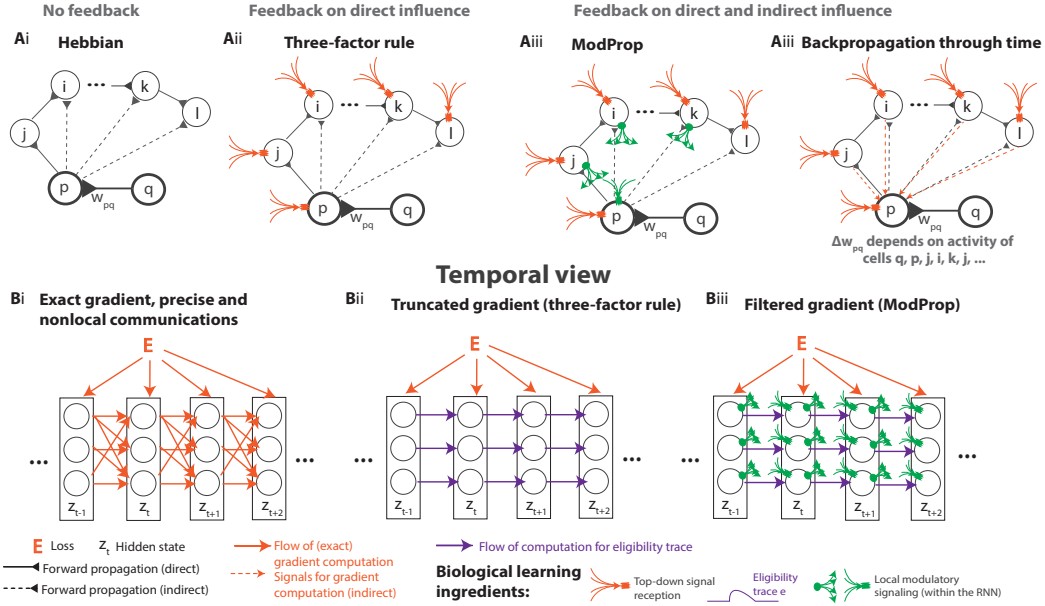

Figure 2: **Local modulatory signaling for gradient estimation.** A) A spatial view of learning rules for updating weight $W_{pq}$. (i) Hebbian learning, where weight update depends only on pre-/post-synaptic activities. (ii) Three-factor learning [4, 5, 55], which updates weights using additional top-down learning signals, severely truncates the exact gradient. (iii) ModProp also accounts for (filtered) distant feedback information delivered through synapse-type-specific neuromodulation; (iv) BPTT computes the exact gradient: weight update involves nonlocal information, i.e. activities of indirectly connected units. B) A temporal view. Bi) BPTT propagates the precise intercellular dependencies in an acausal manner. Bii) Three-factor learning rule neglects all the intercellular dependencies in the temporal propagation of the credit signal. Biii) ModProp **approximates** such spatiotemporal dependencies through local neuromodulatory signals (Eq. 6). ModProp approximates the exact gradient by assuming similar connectivity among cells of the same type, and filtering the indirect effects on loss from neurons that are potentially many synapses away. Figure 6 provides a summary of approximations made by ModProp.

**time-invariant**, a property likely required for biological plausibility. We also define $S$, the arbitrary number of credit propagation steps, and it will become clear later that this corresponds to the number of modulatory filter taps. With this, the estimated gradient becomes:

$$\frac{\mathrm{d}\, E}{\mathrm{d}\, W_{pq}}|_t \approx \frac{\partial E}{\partial z_{p,t}} e_{pq,t} + \sum_j \frac{\partial E}{\partial s_{j,t}} \sum_{s=1}^{S} (W^s)_{jp}\, \mu^{s-1}\, e_{pq,t-s} \qquad (4)$$

where $e_{pq,t} \approx \frac{dz_{p,t}}{dW_{pq}}$ ($e_{pq,t}$ is defined precisely in Appendix Eq. S24) can be interpreted as a persistent Hebbian "eligibility trace" [8–10] that keeps a fading memory of past coincidental pre- and postsynaptic activity [5]. Here, $(W^s)_{jp}$ represents the $(jp)^{\text{th}}$ entry of $W^s$, $W$ raised to the power of $s$, and $(W^1)_{jp} = W_{jp}$. Details of the derivation can be found in Appendix A.

It is important to note that **Approximation 1** is applied only to the nonlocal gradient terms, i.e. exact activation derivative is used during the computation of the eligibility trace that is local to the pre- and postsynaptic neurons. Also, we treat $\mu$ as a hyperparameter in our simulations and explain how this is tuned in Appendix F. One should note that tuning the hyperparameter $\mu$ properly is important, because the estimated gradient can explode when $\mu$ is too large and ModProp approaches the three-factor rule when $\mu$ is too small. Future work involves improving ModProp with an adaptive $\mu$ for better numerical stability and accuracy.

**Approximation 2** replaces **synapse-specific** feedback weights with **type-specific** weights:

$$(W^s)_{jp} \approx (W^s)_{\alpha\beta} \quad \text{(approximation 2)}. \tag{5}$$

Here, cell $j$ belongs to type $\alpha$, cell $p$ is of type $\beta$ and $C$ denotes the set of cell types. (e.g., $W_{\alpha\beta} = \mathbb{E}_{j\in\alpha, p\in\beta}[W_{jp}], \alpha, \beta \in C$.) This approximation is due to the type-specific nature of modulatory channels [23]. We call these modulatory weights synapse-type-specific (as opposed to cell-type-specific) to emphasize the connectivity-based grouping. Details of how these modulatory weights were obtained can be found in Appendix A.

**ModProp: filtering credit signals via local neuromodulation:** Substituting **Approximation 1** and **Approximation 2** into Eq. S4 and 2 leads to the ModProp update:

$$\Delta W_{pq}|_{ModProp} \propto L_p \times e_{pq} + \left( \sum_{\alpha\in C} \left( \sum_{j\in\alpha} L_j \text{ activity}_j \right) \times F_{\alpha\beta} \right) * e_{pq},$$

$$F_{\alpha\beta,s} = \mu^{s-1}(W^s)_{\alpha\beta}, \tag{6}$$

where $L$ and $e$ denote top-down learning signal and eligibility trace, respectively. We postulate that $F_{\alpha\beta}$ represents type-specific filter taps of GPCRs expressed by cells of type $\beta$ to precursors secreted by cells of type $\alpha$; $*$ is the convolution operation with $S$ as the number of filter taps. Note that the matrix powers $(W^s)_{\alpha\beta}$ appearing in the values of different filter taps $F_{\alpha\beta,s}$ could be genetically pre-determined as part of the cell identity and optimized over evolutionary time scales. (Appendix Figure S1 shows successful learning using fixed modulatory weights.) Details of a biological interpretation of Eq. 6 can be found in Appendix D.1. Briefly, the secretion of top-down (TD) learning signals can selectively activate a biochemical process at the post-synaptic neuron, which can then act as a temporal filter on the eligibility trace.

Observing that neurons of the same type demonstrate consistent properties across a range of features (e.g. connectivity, physiology, gene expression) [23, 57], we use two cell types with consistent wiring, neuromodulation, and type of synaptic action (excitatory/inhibitory) in our relatively simple models.

In summary, we propose a synaptic weight update rule, where the eligibility trace is compounded not only with top-down learning signals – as in modern biologically-plausible learning rules [7, 8] – but also with local modulatory pathways through convolution (Figure 1). This modulatory mechanism allows the propagation of credit signals through an arbitrary number of recurrent steps.

## 3.2 Properties of ModProp

**Remark.** *The biologically plausible implementation (Eq. 6) of ModProp complexity scales as $O(SN^2)$ per time step $t$.*

Here, $N$ and $S$ denote the number of recurrent units and the number of filter taps. As seen in Eq. 4, the number of filter taps corresponds to the number of recurrent steps for which the credit information is propagated. We also present an alternative implementation with potentially lower cost later in Proposition 1. Details for cost analysis and biological interpretation can be found in Appendix D.1.

We next show through Theorem 1 that learning with synapse-type-specific weights leads to loss reduction at every step on average. The mechanistic intuition behind Theorem 1 is that, in the presence of a statistical connection between synaptic weights (forward path) and modulatory weights (feedback path), the modulatory feedback signal each neuron receives can be a good estimate of how its activity contributes to the overall task error, which can be used for effective tuning of incoming synaptic weights to reduce the error. Figure S5 compares the angle with the exact gradient across different learning algorithms to demonstrate that the direction of the approximate gradient computed by ModProp is similar to (aligned with) the direction of the exact gradient, thereby reducing the loss.

In the following, we define the "residual weights" away from the cell-type averages as $\epsilon_{ij} := W_{ij} - W_{\alpha\beta}$ and $(\epsilon_s)_{ij} := (W^s)_{ij} - (W^s)_{\alpha\beta}$, where $i \in \alpha, j \in \beta, \alpha, \beta \in C$. We consider these terms as stochastic so that the circuit output and the eligibility traces are also stochastic, as functions of $\epsilon_s$. We consider the cell type averages and the output connection strengths as deterministic. (Merely modifying the uncorrelatedness assumption below would extend our results to stochastic output connection strengths.) Below, $\mathbb{E}$ is short for $\mathbb{E}_\epsilon$, $\Delta E|_t$ denotes the change in the task error at time $t$ and $\Delta E|_{pq,t}$ denotes the contribution of the synapse $(pq)$ to it.

**Theorem 1.** *Consider linear RNNs $\mathcal{N}$, $\widehat{\mathcal{N}}$ with weight matrices $W$, $\widehat{W}$, respectively and identical architectures. Let $\widehat{W}_{ij} = W_{\alpha\beta}$, $\forall i \in \alpha, j \in \beta$. Assume a small enough learning rate such that the remainder of the first-order Taylor expansion of loss is negligible. For network $\mathcal{N}$, if $\mathbb{E}[\epsilon_{ij}] = 0$, $(\epsilon_s)_{ik}$ and $\epsilon_{kj}$ are uncorrelated for $i \neq j$, and $\epsilon_s$ and $(y_t - y_t^*)^2 e_{pq,t'} e_{pq,t-s}$ are uncorrelated for any $s, t, t'$, then $\mathbb{E}[\Delta E|_{pq,t}] \leq 0$ and $\mathbb{E}[\Delta E|_t] \leq 0$. Moreover, $\mathbb{E}[\Delta E|_t] < 0$ if gradient descent is possible for network $\widehat{\mathcal{N}}$. (Proof is in Appendix E.)*

Note that the uncorrelatedness assumption above is relatively mild because any single connection strength is typically a very poor predictor of network activity, especially as the network size grows. Note also that gradient update can introduce a drift in residual weights, requiring a similar drift in cell-type averages for the strict inequality to be applicable over multiple update steps.

## 3.3 Simulation results

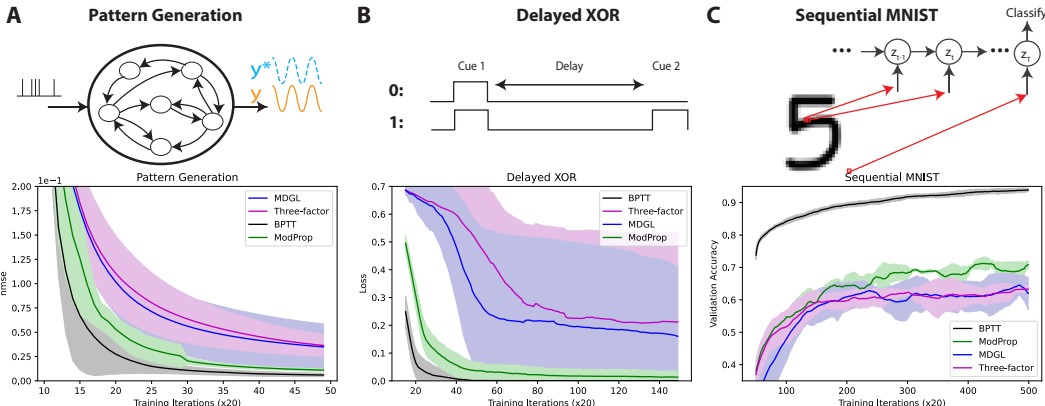

Figure 3: **Modulatory signaling of credit information on long-term recurrent interactions can improve learning outcomes.** ModProp improves the learning performance over existing bio-plausible rules. This experiment examines the performance due to Approximation 1 (Eq. 3) before any cell-type approximation of modulatory weights (Eq. 5). A) Learning to produce a time-resolved target output pattern. B) A delayed XOR task, where the network determines if two cue alternatives — the presence or absence of input represented by $1$ or $0$ — match or mismatch after a delay, requiring memory via recurrent activity. c) Pixel-by-pixel MNIST task [58]. Note that this task is unlikely to be solved effectively by humans. (See text.) Consistent with the original MDGL paper [6], we also find that MDGL confers little advantage over the three-factor rule (e-prop) under dense connectivity. Solid lines/shaded regions: mean/standard deviation of loss curves across five runs.

To test the ModProp formulation, we study its performance in well-known tasks involving temporal processing: pattern generation, delayed XOR, and sequential MNIST. We compare the learning performance of ModProp with the state-of-the-art biologically-plausible learning rules (MDGL [6] and e-prop [5] (labeled as "three-factor")), as well as BPTT to provide a lower bound on task error. Note, BPTT and RTRL both compute the exact gradient, so they should be identical in terms of performance. Here, we chose BPTT due to computational efficiency. Consistent with [6], we also found MDGL to confer little advantage over e-prop when all neurons are connected to the readout (Figure 3), so we focus our analysis on that case.

We first study pattern generation with RNNs, where the aim is to produce a one-dimensional target output, generated from the sum of five sinusoids, given a fixed Gaussian input realization. We change the target output and the random input along with initial weights for different training runs, and show the learning curve in Figure 3A across seven such runs. By using a densely connected network, we observe that MDGL confers little advantage over the three-factor rule (e-prop), as reported in the original paper [6]. Moreover, communicating longer indirect effects, despite being filtered, leads to improved learning outcomes, as demonstrated by the superior performance of ModProp over MDGL.

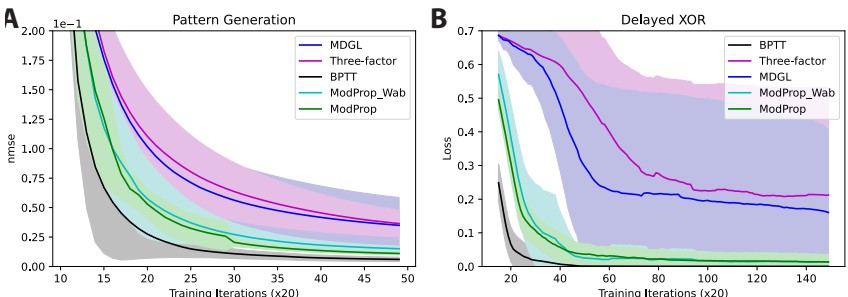

Figure 4: **Efficient learning with type-specific modulatory weights.** In addition to Approximation 1 studied in Figure 3, this figure investigates the effect of Approximation 2 (labeled as ModProp_Wab), which uses type-specific, rather than synapse-specific feedback weights for signaling credit information (Eq. 5). Here, ModProp_Wab uses only two modulatory types mapped to the two main cell classes. The cell type approximation does not result in any significant performance degradation in A) the pattern generation task and B) the delayed XOR task. This analysis is not done for the sequential MNIST task, where neurons were not divided into E and I types. Solid lines/shaded regions: mean/standard deviation of loss curves across five runs.

Next, to study how RNNs learn to process discrete cues that impact delayed outcomes, we consider a delayed XOR task: two cue alternatives, 1 or 0, are encoded by the presence/absence of input. The network is trained to remember the first cue and learn to compare it with the second cue delivered at a later time to determine if the two cues match. Figure 3B illustrates the learning curves for this task. We observe the same general conclusion as the previous task. Some learning curves have a standard deviation, which indicates that the network struggled to learn the task with these rules for some seeds; based on the standard deviation of the curves, it seems possible for ModProp to perform similarly as three-factor for some seeds, but the focus here is to examine performance across many runs. In Appendix Figure S3, we also examined the learning performance of the same set of learning rules for a longer delay period. Interestingly, the performance of ModProp approaches that of BPTT for this task and the previous one, further closing the gap between artificial network training and biological learning mechanisms in performing credit assignment over a long period.

Finally, we study the pixel-by-pixel MNIST [58] task, which is a popular machine learning benchmark. Although it is not a task that the brain would solve well (i.e., humans would struggle to predict the digit with only one pixel presented at a time), we investigate it to test the limits of spatiotemporally filtered credit signals for tasks that demand temporally precise input integration. Figure 3C illustrates the learning curves for this task. While the performance ordering of learning rules is still the same as in previous tasks, we observe a wider gap between ModProp and BPTT. Since the time-invariant filter approximation (Eq 3) restricts the spatiotemporal resolution of the credit signal, this is in line with our expectation that ModProp will struggle with tasks that demand highly precise spatiotemporal integration, such as the pixel-by-pixel MNIST task that even the brain would struggle to solve.

As a proof-of-concept study, we initially focused our analysis on the approximation performance by imposing time-invariant filter taps (Eq. 3). Next, in Figure 4, we investigate the learning performance using type-specific rather than synapse-specific feedback modulatory weights (Eq. 5). These type-specific weights were calculated using weight averages as in [6]. Later, we repeat these simulations using fixed random modulatory weights in Appendix Figure S1; we also demonstrate the superior performance of ModProp with fixed random modulatory weights for the "copy task" [59] in Appendix Figure S4. Little performance degradation is observed for only two modulatory types mapped onto the two main cell classes, indicating the effectiveness of cell-type discretization. Other than the sequential MNIST task, cells are divided into two main cell classes — with $80\%$ of the cells being excitatory (E) and $20\%$ being inhibitory (I) — and obeyed connection sign constraints.

Lastly, we demonstrate the advantage of our weight update rule in an online learning setting of the pattern generation task, where weights are updated in real time (Figure 5). For that, we also derive a cost and storage efficient in silico implementation of ModProp in Proposition 1.

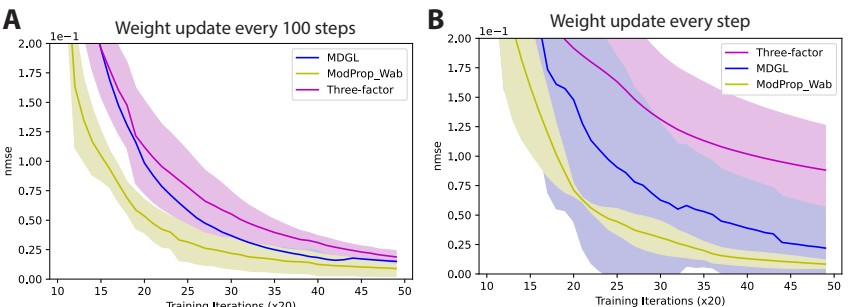

Figure 5: **Superior performance of ModProp in an online learning setting**. We investigate an online learning version of the pattern generation task, where weights are updated either A) every 100 time steps or B) every single step. Here, ModProp uses the efficient online learning implementation derived in Proposition 1. Plotting conventions follow those of previous figures.

**Proposition 1.** *ModProp has an online in-silico (not necessarily biologically-plausible) implementation with $O(CN^2)$ storage and $O(C^2N^2)$ computational complexity, where $C$ is the number of cell types. (Proof is in Appendix D.2.)*

## 4 Discussion

A central question in the study of biological and artificial intelligence is how the temporal credit assignment problem is solved in neuronal circuits [3]. Motivated by recent genomic evidence on the widespread presence of local modulatory networks [23, 24], we demonstrated how such signaling could complement Hebbian learning and global neuromodulation (e.g., via dopamine) to achieve biologically plausible propagation of the credit signal through arbitrary time steps. Going beyond the scalar feedback provided by global top-down modulation, our study proposes how detailed vector feedback information [1] can be delivered in neuronal circuits. Instead of the severe temporal truncations of the gradient information proposed by the state-of-the-art [4–6], ModProp offers a framework where the full temporal feedback signal can be received, albeit via low-pass filtering at the post-synaptic neuron due to specificity at the level of neuronal types, and not individual neurons. Moreover, predictions generated by ModProp on the role of a family of signaling molecules (e.g. neuropeptides) could potentially be tested experimentally: physiology of multiple individual cells can be monitored in modern neurobiology experiments [25, 60]. Blocking certain peptidergic receptors of the neurons that are involved with learning a task and comparing the performance to that without blocking can provide a test for the role of peptidergic communication.

While feedback alignment [44] addresses the weight transport problem in feedforward networks, it is not clear in RNNs which biological pathways would implement **temporal** feedback. Our model suggests that such pathways could come from synapse-type-specific local neuromodulation. In addition to improved performance compared to the state-of-the-art across all of our simulations, our theoretical and experimental results show that ModProp can be implemented efficiently for online learning. Figure 6 briefly mentions some of the connections of ModProp to feedback alignment. Together, these findings suggest that synapse-type-specific local modulatory signaling could be a neural mechanism for propagating credit information over more than just a few recurrent steps.

Among the many future directions, a natural extension could be to investigate the performance of ModProp across a broader range of tasks. This could include situations where the assumptions in deriving the rule (e.g., stationarity of activity) are severely violated. This might improve the approximations introduced here. Similarly, we observed a significant gap between ModProp and BPTT for the pixel-by-pixel MNIST task that demands precise temporal integration of input (Figure 3C), but we discussed earlier that this is a challenging task for the brain. Additionally, this study focuses on dense networks with ReLU activation; future directions include investigating two biologically relevant paradigms: sparse connectivity and a diverse set of activation functions (spike-based in particular). Although ModProp can be applied in theory to temporal credit assignment over an arbitrary duration, the presented learning rule accounts for dependencies at every single step (as for BPTT). This means that, similar to BPTT, it is ill-suited for very long-term credit assignment [61]. An interesting line of

research attends to the issue of extracting relevant, rather than full, memories [61, 62]. Investigating how our approximations could potentially be combined with memory sparsification techniques to perform very long-term biologically-plausible credit assignment can be fruitful [61].

While our paper advances the basic science of learning and we do not foresee immediate societal impacts of our work, its benefits to both neuroscience and deep learning research could have long-term (positive or negative) societal impacts. It is hard to overstate the philosophical implications of understanding how the brain works (and, in particular, learns). Moreover, such understanding could guide us toward curing certain diseases of the brain. Nevertheless, in the same vein, such future tools could also enable abuse if left unregulated. On the machine learning side, our method can be considered a low-cost alternative to the ubiquitous BPTT algorithm. In this sense, our algorithm or a future method that builds on it can make tasks that are currently approachable by just a few wealthy entities available to many more practitioners. On the flip side, as with many other capable machine learning tools, such entities are also capable of utilizing low-cost learning to conquer even harder tasks, which is potentially a reason for concern. Finally, data-driven tools will ultimately reflect the various biases in their training data and our method is no exception.

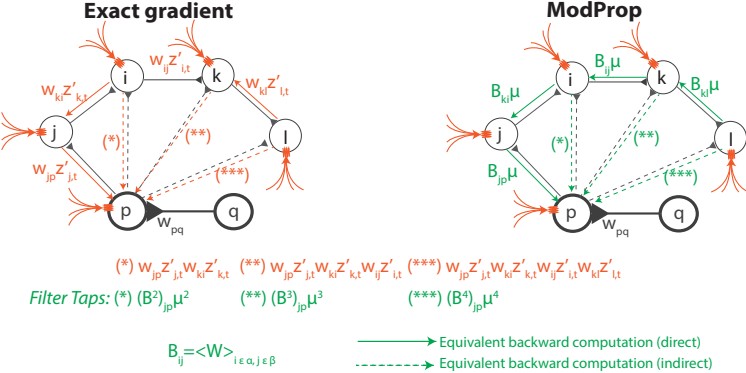

Figure 6: **ModProp brings two approximations to the nonlocal gradient terms.** First, ModProp uses type-specific feedback weights (modulatory weights), rather than cell-specific feedback weights (Eq. 3). Second, ModProp approximates the activation derivative (Eq. 5). Similar to feedback alignment for feedforward networks [44], different weights are used during the backward pass than the forward pass due to the well-known weight transport problem. This, however, won't be adequate for RNN settings. On top of the type-specific feedback weights approximation, time-invariant activation derivative approximation was also applied for time-invariant filtering (as explained in the texts surrounding Eq. 4 and Eq. 6). Any unspecified symbols in the illustration were defined in Figure 2.

## Acknowledgements

We thank the Allen Institute founder, Paul G Allen, for his vision, encouragement and support. YHL is supported by the NSERC PGS-D program. This work was facilitated through the use of the UW Hyak supercomputer system.

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
