# A  Methods

## A.1  Network Model

We consider a discrete-time implementation of a rate-based recurrent neural network (RNN) similar to the form in [56]. We denote the observable states, i.e. firing rates, as $z_t$ at time $t$, and the corresponding internal states as $s_t$. The dynamics of those states are governed by

$$s_{j,t+1} = \eta \, s_{j,t} + (1 - \eta) \left( \sum_{l \neq j} W_{jl} \, z_{l,t} + \sum_p W_{jm}^{\text{IN}} \, x_{m,t+1} \right)$$
$$z_{j,t} = ReLU(s_{j,t}), \tag{S1}$$

where $\eta = e^{-dt/\tau_m}$ denotes the leak factor for simulation time step $dt$ and membrane time constant $\tau_m$, $W_{lj}$ denotes the weight of the synaptic connection from neuron $j$ to $l$, $W_{jm}^{\text{IN}}$ denotes the strength of the connection between the $m^{th}$ external input and neuron $j$ and $x_t$ denotes the external input at time $t$. Threshold adaptation is not used here in order to focus on capacity of the temporal credit propagation mechanism. We focused on ReLU activation due to its wide adoption in both deep learning and computational neuroscience communities; as discussed, we leave extension to other activation functions (spike-based in particular) for future work.

The readout $y$ is a linear transformation of the hidden state

$$y_{k,t} = \sum_j W_{kj}^{\text{OUT}} z_{j,t} + b_k^{\text{OUT}}, \tag{S2}$$

where $W_{kj}^{\text{OUT}}$ denotes the strength of the connection from neuron $j$ to output neuron $k$, $b_k^{\text{OUT}}$ denotes the bias of the $k$-th output neuron.

We quantify how well the network output matches the desired target using loss function $E$:

$$E = \begin{cases} \frac{1}{2} \sum_{k,t} (y_{k,t}^* - y_{k,t})^2, & \text{for regression tasks} \\ - \sum_{k,t} \pi_{k,t}^* log \pi_{k,t}, & \text{for classification tasks} \end{cases} \tag{S3}$$

where $y_{k,t}^*$ is the time-dependent target, $\pi_{k,t}^*$ is the one-hot encoded target and $\pi_{k,t} = \text{softmax}_k(y_{1,t}, \ldots, y_{N_{OUT},t}) = \exp(y_{k,t}) / \sum_{k'} \exp(y_{k',t})$ is the predicted category probability.

## A.2  Gradient descent learning in RNNs

**Notation for Derivatives**: There are two types of computational dependencies in RNNs; direct and indirect dependencies. We distinguish direct dependencies versus all dependencies (including indirect ones) using partial derivatives ($\partial$) versus total derivatives ($d$), respectively.

Without loss of generality, consider a function $f(x, y)$, where $y$ itself may depend on $x$. The partial derivative $\partial$ of $f$ at $x_0$ considers $y$ as a constant, and evaluates as $\frac{\partial f(x,y)}{\partial x}|_{x_0, y(x_0)}$; i.e. the derivative calculation only considers how $x$ directly affects $f$. The total derivative d, on the other hand, may not treat $y$ as a constant and evaluates as $\frac{df(x,y)}{dx} = \frac{\partial f(x,y)}{\partial x}|_{x_0, y(x_0)} + \frac{\partial f(x,y)}{\partial y}|_{x_0, y(x_0)} \frac{\partial y}{\partial x}|_{x_0}$; i.e. the derivative calculation also takes into account how $x$ can indirectly affect $f$ through $y$.

As an example in our network, variable $W_{pq}$ can impact state $s_{p,t}$ directly through Eq. S1, i.e. $\frac{\partial s_{p,t}}{W_{pq}} = (1 - \eta) z_{q,t-1}$. On the other hand, $W_{pq}$ can also impact $s_{p,t}$ indirectly through other cells in the network: i.e. the dependence of $s_{p,t}$ on $W_{pq}$ and all $s_{j,t'}$ ($t' < t$, $j \in \{1, ..., N\}$) affected by $W_{pq}$ are taken into account for the derivative calculation, which leads to the recursive equation in Eq. S8.

**Exact gradient computation and locality issue**: In gradient descent learning, all weight parameters (input weights $W^{IN}$, recurrent weights $W$ and output weights $W^{OUT}$) are adjusted iteratively according to the error gradient. This error gradient can be calculated with classical machine learning algorithms, backpropagation through time (BPTT) and real time recurrent learning (RTRL) [42], which uses different factorization but yield equivalent results. However, the BPTT factorization depend on future activity, which poses an obstacle for online learning and biological plausibility. Our learning rule derivation follows the RTRL factorization because it is causal.

RTRL factors the error gradient across time and space as

$$\frac{\mathrm{d}\,E}{\mathrm{d}\,W_{pq}}\Big|_t = \sum_j \frac{\partial E}{\partial s_{j,t}} \frac{\mathrm{d}\,s_{j,t}}{\mathrm{d}\,W_{pq}} \tag{S4}$$

$$\frac{\mathrm{d}\,z_{j,t}}{\mathrm{d}\,W_{pq}} = h_{j,t} \frac{\mathrm{d}\,s_{j,t}}{\mathrm{d}\,W_{pq}}, \text{where } h_{j,t} := \frac{\partial z_{j,t}}{\partial s_{j,t}} \tag{S5}$$

$$\frac{\partial s_{j,t}}{\partial W_{pq}} = \delta_{jp}(1-\eta)z_{q,t-1} \tag{S6}$$

$$\frac{\partial s_{j,t}}{\partial s_{l,t-1}} = \begin{cases} \eta, & j = l \\ \frac{\partial s_{j,t}}{\partial z_{l,t-1}} \frac{\partial z_{l,t-1}}{\partial s_{l,t-1}} = (1-\eta)W_{jl}h_{l,t-1}, & j \neq l \end{cases} \tag{S7}$$

$$\frac{\mathrm{d}\,s_{j,t}}{\mathrm{d}\,W_{pq}} = \frac{\partial s_{j,t}}{\partial W_{pq}} + \sum_l \frac{\partial s_{j,t}}{\partial s_{l,t-1}} \frac{\mathrm{d}\,s_{l,t-1}}{\mathrm{d}\,W_{pq}}$$

$$= \frac{\partial s_{j,t}}{\partial W_{pq}} + \eta \frac{\mathrm{d}\,s_{j,t-1}}{\mathrm{d}\,W_{pq}} + (1-\eta)\underbrace{\sum_{l \neq j} W_{jl} \frac{\partial z_{l,t-1}}{\partial s_{l,t-1}} \frac{\mathrm{d}\,s_{l,t-1}}{\mathrm{d}\,W_{pq}}}_{\text{depends on all weights } W_{jl}}. \tag{S8}$$

following the derivative notation explained above. The factor $\frac{\partial E}{\partial z_{j,t}}$ in Eq. S4 can be interpreted as the top-down learning signal, which is defined as $L_{j,t} := \sum_k W_{kj}^{OUT}(y_{k,t} - y_{k,t}^*)$ for regression tasks [5]. It is straightforward to compute. However, the triple tensor $\frac{\mathrm{d}\,s_{j,t}}{\mathrm{d}\,W_{pq}}$ requires $O(N^3)$ memory and $O(N^4)$ computation costs. It keeps track of all the paths that $z_{j,t}$ can affect $W_{pq}$ (for every $j, p, q$). Moreover, it poses a significant challenge to biological plausibility: updating each weight $W_{pq}$ requires knowing all other weights $W_{jl}$ (for every $j$ and $l$) in the network, and that information should be inaccessible to neural circuits.

To address this, references [4] and [5] dropped the problematic terms so that the updates to weight $W_{pq}$ would only depend on pre- and post-synaptic activity, and applied this truncation to train rate- and spike-based networks, respectively. However, such truncation results in limited performance.

### A.3 Derivation of ModProp

We ask how intercellular neuromodulation might communicate the expensive spatiotemporal dependency in the factor $\frac{\mathrm{d}\,s_{j,t}}{\mathrm{d}\,W_{pq}}$. Along with the interpretation of $L_{j,t} := \frac{\partial E}{\partial z_{j,t}}$ as top-down learning signal, let $e_{pq,t}$ denote the eligibility trace of coincidental activation between presynaptic cell $q$ and postsynaptic cell $p$ [5]. The following derivation leads to our learning rule. The leak factor is omitted in the derivation below ($\eta = 0$) for readability, and we substitute Eq. S8 into Eq. S4 and repeatedly expand the expensive $\frac{\mathrm{d}\,s}{\mathrm{d}\,W}$ factor using Eq. S8:

$$\frac{\mathrm{d}\,E}{\mathrm{d}\,W_{pq}}\Big|_t = \sum_j \frac{\partial E}{\partial z_{j,t}} h_{j,t} \left( \delta_{jp}z_{q,t-1} + \sum_l W_{jl}h_{l,t-1} \frac{\mathrm{d}\,s_{l,t-1}}{\mathrm{d}\,W_{pq}} \right) \tag{S9}$$

$$= \frac{\partial E}{\partial z_{p,t}} h_{p,t} z_{q,t-1} + \sum_j \frac{\partial E}{\partial z_{j,t}} h_{j,t} \sum_l W_{jl}h_{l,t-1} \frac{\mathrm{d}\,s_{l,t-1}}{\mathrm{d}\,W_{pq}} \tag{S10}$$

$$= \frac{\partial E}{\partial z_{p,t}} e_{pq,t} + \sum_j \frac{\partial E}{\partial z_{j,t}} h_{j,t} \sum_l W_{jl}h_{l,t-1} \left( \delta_{lp}z_{q,t-2} + \sum_k W_{lk}h_{k,t-2} \frac{\mathrm{d}\,s_{k,t-2}}{\mathrm{d}\,W_{pq}} \right) \tag{S11}$$

$$= \dots$$

$$\frac{\mathrm{d}\,E}{\mathrm{d}\,W_{pq}}\Big|_t = \frac{\partial E}{\partial z_{p,t}} e_{pq,t} + \left( \sum_j \frac{\partial E}{\partial z_{j,t}} h_{j,t} W_{jp} \right) e_{pq,t-1}+$$

$$\sum_j \frac{\partial E}{\partial z_{j,t}} h_{j,t} \sum_{s=2}^{S} \sum_{i_1,\ldots,i_{s-1}} W_{ji_1} W_{i_1 i_2} \ldots W_{i_{s-1}p} h_{i_1,t-1} \ldots h_{i_{s-1},t-s+1} e_{pq,t-s} \quad \text{(S12)}$$

$$\stackrel{(a)}{\approx} \frac{\partial E}{\partial z_{p,t}} e_{pq,t} + \left( \sum_j \frac{\partial E}{\partial z_{j,t}} h_{j,t} W_{jp} \right) e_{pq,t-1}$$

$$+ \sum_j \frac{\partial E}{\partial z_{j,t}} h_{j,t} \sum_{s=2}^{S} (W^s)_{jp}\, e_{pq,t-s} \frac{1}{N^{s-1}} \sum_{i_1,\ldots,i_{s-1}} h_{i_1,t-1} h_{i_2,t-2} \ldots h_{i_{s-1},t-s+1} \quad \text{(S13)}$$

$$= \frac{\partial E}{\partial z_{p,t}} e_{pq,t} + \sum_j \frac{\partial E}{\partial z_{j,t}} h_{j,t} \sum_{s=1}^{S} (W^s)_{jp}\, H(t,s)\, e_{pq,t-s}, \quad \text{(S14)}$$

where $H(t,1) = 1$, $H(t,s) = \frac{1}{N^{s-1}} \sum_{i_1,\ldots,i_{s-1}} h_{i_1,t-1} h_{i_2,t-2} \ldots h_{i_{s-1},t-s+1}$ for $s = 1,\ldots,S$ and $S$, as explained later, is the number of filter taps. Again, we neglected the leak factor in the derivation for readability but included in the actual simulations. The only approximation step above, $(a)$, is made by using a point estimate assuming that the $W$ and $h$ chains are uncorrelated and the central limit theorem applies. We note that in a linear network, all the activation derivatives $h$ would be 1, making the approximation exact.

We expand on our explanation for step $(a)$ approximation. We define a $W$-chain (of length $l$) as

$$\prod_{\phi=1}^{l} W_{i_\phi i_{\phi+1}}, \quad \text{(S15)}$$

for any indices $i_1,\ldots,i_{l+1} \in \{1,...,N\}$. Similarly, we define an $h$-chain (of length $l'$) as

$$\prod_{\theta=1}^{l'} h_{j_\theta,t-\theta}, \quad \text{(S16)}$$

for any indices $j_1,\ldots,j_{l'} \in \{1,...,N\}$. With these definitions, we call the $W$-chain $W_{i_1,\ldots,i_{s-1}} = W_{ji_1} W_{i_1 i_2} \ldots W_{i_{s-1}p}$ and the $h$-chain $h_{i_1,\ldots,i_{s-1}} = h_{i_1,t-1} \ldots h_{i_{s-1},t-s+1}$ uncorrelated if

$$\mathbb{E}_{i_1,\ldots,i_{s-1}} W_{i_1,\ldots,i_{s-1}} h_{i_1,\ldots,i_{s-1}} = \mathbb{E}_{i_1,\ldots,i_{s-1}} W_{i_1,\ldots,i_{s-1}} \mathbb{E}_{i_1,\ldots,i_{s-1}} h_{i_1,\ldots,i_{s-1}}. \quad \text{(S17)}$$

Considering $W_{i_1,\ldots,i_{s-1}}$ and $h_{i_1,\ldots,i_{s-1}}$ as random i.i.d. samples indexed by $i_1,\ldots,i_{s-1}$, the central limit theorem states that

$$\sum_{i_1,\ldots,i_{s-1}} W_{i_1,\ldots,i_{s-1}} h_{i_1,\ldots,i_{s-1}} \sim \mathcal{N}(N^{s-1}\mathbb{E}[W_{i_1,\ldots,i_{s-1}} h_{i_1,\ldots,i_{s-1}}], N^{s-1}\mathrm{Var}(W_{i_1,\ldots,i_{s-1}} h_{i_1,\ldots,i_{s-1}}))$$
$$\text{(S18)}$$

as the sum tends to infinity. Here, we simply use the i.i.d. assumption even though stronger versions of the Central Limit Theorem need weaker assumptions than i.i.d. When the $W$- and $h$-chains are uncorrelated, we take the mean of this distribution as a point estimate (note, however, the growing variance) to arrive at the following approximation:

$$\sum_{i_1,\ldots,i_{s-1}} W_{i_1,\ldots,i_{s-1}} h_{i_1,\ldots,i_{s-1}} \approx N^{s-1}\mathbb{E}W_{i_1,\ldots,i_{s-1}} h_{i_1,\ldots,i_{s-1}} = N^{s-1}\mathbb{E}W_{i_1,\ldots,i_{s-1}} \mathbb{E}h_{i_1,\ldots,i_{s-1}}.$$
$$\text{(S19)}$$

Since $\mathbb{E}W_{i_1,\ldots,i_{s-1}} = \frac{1}{N^{s-1}}(W^s)_{jp}$ (by the same application of central limit theorem as above) when $i_1,\ldots,i_{s-1}$ are distributed uniformly over valid index ranges, we conclude that

$$\sum_{i_1,\ldots,i_{s-1}} W_{i_1,\ldots,i_{s-1}} h_{i_1,\ldots,i_{s-1}} \approx (W^s)_{jp} \frac{1}{N^{s-1}} \sum_{i_1,\ldots,i_{s-1}} h_{i_1,\ldots,i_{s-1}}, \quad \text{(S20)}$$

where the expectation of the $h$-chain is replaced by its empirical estimate.

To put this derivation in terms of biological components, we make the following further approximations. First, we link modulatory weights to type-specific GPCR efficacies, which means they are type-specific, i.e. $(W^s)_{jp} \approx (W^s)_{\alpha\beta}$, for type-indices $\alpha$ and $\beta$ in a set of possible classes $C$. Second, $H(t, s)$ should be time-invariant, i.e. $H(t, s) = H(s)$, since biological filter properties should not vary rapidly across time.

Interestingly, we observe that $H(s)$ is an average of activation history across time and cells (Eq. S14). In particular, when the activation function is ReLU, one can think of $H(s)$ as approximating the number of activation chains with length $s$ (divided by the total number of possible chains). Thus, a crude starting point would be to assume first-order stationarity, i.e., assume the average activity level remains invariant ($\frac{1}{N}\sum_i h_{i,t} := \mu_t \approx \mu, \forall t$). Then

$$
\begin{aligned}
H(t, s) &= \frac{1}{N^{s-1}} \sum_{i_1,\ldots,i_{s-1}} h_{i_1,t-1} \ldots h_{i_{s-1},t-s+1} \\
&\stackrel{(a)}{=} \frac{1}{N} \sum_{i_1} h_{i_1,t-1} \frac{1}{N} \sum_{i_2} h_{i_2,t-2} \cdots \frac{1}{N} \sum_{i_{s-1}} h_{i_{s-1},t-s+1} \\
&\approx \mu^{s-1},
\end{aligned}
\tag{S21}
$$

where $\mu$ is a scalar constant and represents global average neuron activation. $(a)$ is because in the case of ReLU activation (where activation derivative $h$ is binary), total number of different activation chain combinations, $\sum_{i_1,\ldots,i_{s-1}} h_{i_1,t-1} \ldots h_{i_{s-1},t-s+1}$, would equal to the product of number of activation at each time step, $\sum_{i_1} h_{i_1,t-1} \sum_{i_2} h_{i_2,t-2} \cdots \sum_{i_{s-1}} h_{i_{s-1},t-s+1}$. This is because to choose a chain of activated neurons from all possible combinations, the number of possible indices to choose from for each step is equal to the number of activated neuron at that step. Indeed, intercellular signaling level is activity-dependent [26]. For implementation, $\mu$ can be treated as a hyperparameter or adapted on a separate timescale. It is also important to note that this activation derivative approximation is only applied to the term (in the equation below) additional to the e-prop term, $\frac{\partial E}{\partial z_{p,t}} e_{pq,t}$, for which the exact activation derivative is still used.

By substituting these further approximations into Eq. S14, the approximated gradient becomes:

$$
\frac{\mathrm{d}E}{\mathrm{d}W_{pq}}\Big|_t \approx \frac{\partial E}{\partial z_{p,t}} e_{pq,t} + \sum_{\alpha \in C} \left( \sum_{j \in \alpha} \frac{\partial E}{\partial z_{j,t}} h_{j,t} \right) \sum_{s=1}^{S} (W^s)_{\alpha\beta}\, \mu^{s-1}\, e_{pq,t-s},
\tag{S22}
$$

and this leads to the ModProp update:

$$
\begin{aligned}
\Delta W_{pq}\big|_{ModProp} &\propto L_p \times e_{pq} + \left( \sum_{\alpha \in C} \left( \sum_{j \in \alpha} \mathrm{L}_j h_j \right) \times F_{\alpha\beta} \right) * \mathrm{e}_{pq}, \\
F_{\alpha\beta,s} &= \mu^{s-1}(W^s)_{\alpha\beta},
\end{aligned}
\tag{S23}
$$

where cell $j$ is of type $\alpha$, cell $p$ is of type $\beta$ and $C$ denotes the set of cell types. $L$ and $e$ denote top-down learning signal and eligibility trace, respectively. Again, activation derivative $h_j$ is closely linked to activity level of neuron $j$. $F$ represents the modulatory filter, $F_{\alpha\beta} * \mathrm{e}_{pq} = \sum_{s=1}^{S} F_{\alpha\beta,s} \mathrm{e}_{pq,t-s}$ is the convolution operation with $S$ as the number of filter taps, and scaling factor $\mu$ is a hyperparameter. For calculating the modulatory weights, the weights were calculated using matrix powers for $s > 1$. (See beginning of Theorem 1 proof.) For $s = 1$, we first examined $W_{\alpha\beta}^1 = <W_{jp}^1>_{j \in \alpha, p \in \beta}$ in the main text. This assumes modulatory weights and synaptic weights co-adapt throughout training and to what extent they co-adapt in neural circuits is unclear. Thus, we also set modulatory weights to fixed random type-specific values and demonstrate the resulting learning performance in Appendix Figure S1. These fixed random type-specific modulatory weights were generated randomly from the distribution of averages of random initial weights. Thus, these fixed random type-specific modulatory weights would be close to the initial synaptic weight averages and could stay close depending on how much these synaptic weight averages change throughout training.

**Eligibility trace implementation**: We here explain the implementation of eligibility trace $e_{pq,t}$:

$$e_{pq,t} := \frac{\partial z_{p,t}}{\partial s_{p,t}} \epsilon_{pq,t}, \tag{S24}$$

$$\epsilon_{pq,t} = \frac{\partial s_{p,t}}{\partial w_{pq}} + \frac{\partial s_{p,t}}{\partial s_{p,t-1}} \epsilon_{pq,t-1}, \tag{S25}$$

which tracks the coincidence of postsynaptic activity $h_{p,t} = \frac{\partial z_{p,t}}{\partial s_{p,t}}$ and a low pass filtering of presynaptic activity stored in $\epsilon_{pq,t}$ ($\frac{\partial s_{p,t}}{\partial w_{pq}} = z_{q,t-1}$ and $\frac{\partial s_{p,t+1}}{\partial s_{p,t}} = \eta$ following Eq. S1). Reference [5] provides a comprehensive discussion on how eligibility traces can be interpreted as derivatives.

# B  Additional simulations

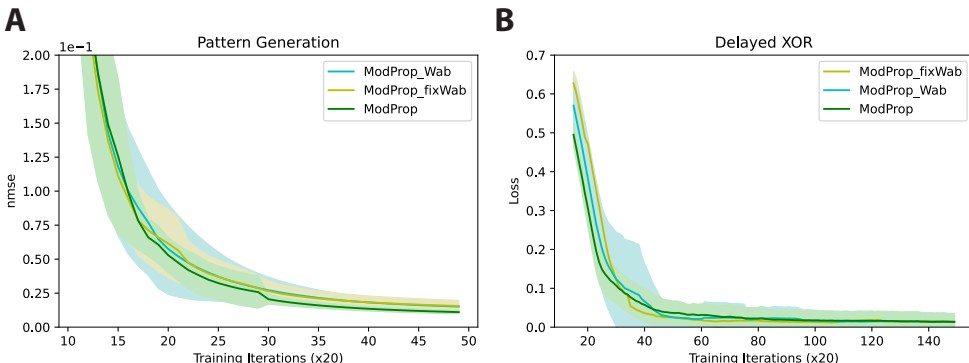

Figure S1: **Effective learning is achievable with fixed random synapse-type-specific modulatory weights.** Figure 4 computes type-specific modulatory weights by averaging forward weight entries in the corresponding pre- and postsynaptic cell group. This assumes that modulatory weights co-adapt with synaptic weights. To what extent they are linked in the brain is unclear. Thus, to test the generality of our learning rule, we re-train using fixed random type-specific modulatory weights and show that leads to negligible performance degradation. Note, sequential MNIST task is not considered in figures that involve synapse-type-specific modulatory weights, as cell types were not considered in that task. Plotting convention follows that of previous figures.

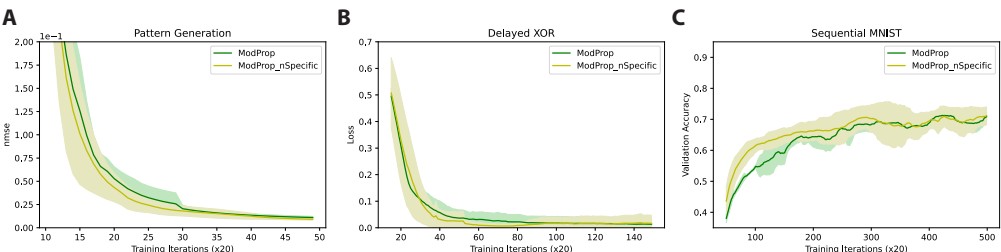

Figure S2: **Restoring neuron specificity in the activation derivative does not lead to significant improvements.** Here, ModProp_global is the basic form of ModProp investigated in the main text, where the activation derivative exhibited no spatiotemporal specificity. ModProp_nSpecific (Eq. S26) takes into account the neuron specificity of the activation derivative and only averages across time steps. This comparison is done for the A) pattern generation task, B) delayed XOR task and C) sequential MNIST task. Plotting convention follows that of previous figures.

We discussed how the basic form of ModProp completely neglects any spatiotemporal specificity in the activation derivative. We ask how much performance gain could we get if we lose only temporal specificity, i.e. only average activation derivative across time points. This would see different neurons as having different average activity. To put this more concretely, we approximate the corresponding factor in Eq. S12 as

$$\sum_{i_1,\ldots,i_{s-1}} W_{ji_1}h_{i_1,t-1}W_{i_1i_2}h_{i_2,t-2}\ldots W_{i_{s-1}p}h_{i_{s-1},t-s+1}$$
$$\approx W_{ji_1}\overline{h_{i_1}}W_{i_1i_2}\overline{h_{i_2}}\ldots W_{i_{s-1}p}\overline{h_{i_{s-1}}} = (\overline{W})^s_{jp}, \tag{S26}$$

where $\overline{W} := W \odot \overline{h}$ with $\overline{h}$ — a $1 - by - N$ vector with each entry corresponding to a neuron-specific mean activation — broadcasted for the element-wise multiplication with $W$. In other words, this restores spatial specificity and the only approximation being made here is to remove temporal specificity of activation derivative. As a practical note, by the famous AM-GM inequality, the estimation ($\Pi_s h_s \approx \overline{h}$) would yield an upper bound of the actual. Thus, we multiply a dampening

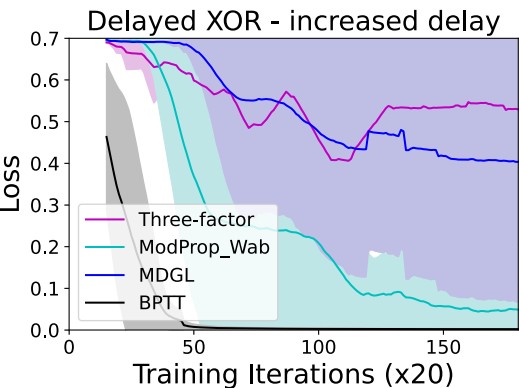

Figure S3: **Delayed XOR task with a longer delay period.** We simulate the delayed XOR task with 1.5 times the delay period used in Figure 3B. Although ModProp (with cell-type approximation) still outperforms other bio-plausible learning rules, the performance degrades (compared to ModProp_Wab in Figure 4B). Moreover, we found all rules (including BPTT) struggle to learn if we increased the delay period to twice of that in Figure 3 without changing other task or network parameters (e.g. cue width and intensity). This connects nicely to our discussion point on the limitation of ModProp in addressing very long temporal credit assignment problems in the absence of a long-term memory mechanism. Solid lines/shaded regions: mean/standard deviation of loss curves across five runs.

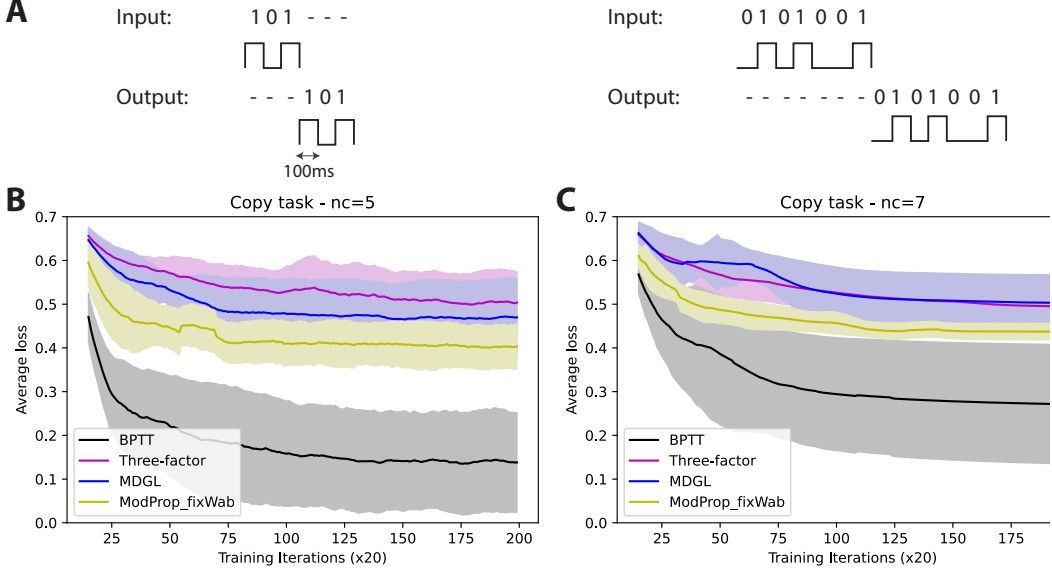

Figure S4: **Copy task with fixed random synapse-type-specific modulatory weights.** Sequences of binary cues are presented to an RNN. For each sequence, once the full sequence has been presented, the network should output the original sequence (with the same value and duration) without any further information [59]. **A)** Examples of input/output pairs at different sequence lengths. Instead of having each cue lasting just 1 step, we have each cue lasting 100 steps (100ms) to mimic the duration of a quick cue flash in biological settings. Superior performance of ModProp even with fixed and random modulatory weights (compared to other biologically plausible rules) is demonstrated for the copy task with a sequence length of **B)** five cues ($nc = 5$) and **C)** seven cues ($nc = 7$). Average loss denotes the binary cross entropy loss computed on target and actual output averaged across time steps. Solid lines/shaded regions: mean/standard deviation of loss curves across five runs.

factor $\mu$ to every $\overline{W}$ for stability, and treat $\mu$ as a hyperparameter. We name this variant of ModProp as "ModProp_nSpecific", and the most basic form we investigated in the main text as "ModProp_global".

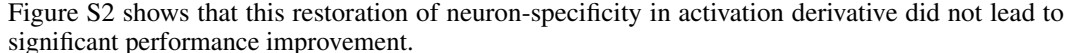

Figure S2 shows that this restoration of neuron-specificity in activation derivative did not lead to significant performance improvement.

## C Further discussion on related algorithms

For efficient online learning in RNNs, approximations to RTRL have been proposed [4, 17, 50–54]. For biological realism (use only local information for local updates) and reduced computational cost, **e-prop** [5] and **RFLO** [4] proposes severe truncation such that the weight update would only depend on pre- and postsynptic neuron activity as well as a top-down error signal that only tells how a neuron directly contributes to the overall network outcome. **MDGL** [6, 63] proposes a less severe truncation than e-prop and RFLO, but it only addresses the contributions to the task error of neurons that are at most 2 synapses away (note, Ref [6] can be considered as a special case of our work, where the filter length is constrained to a single tap). ModProp shows a way of removing this significant limitation and enables the communication and calculation of credit from neurons that can be arbitrarily many synapses away. It also experimentally demonstrates the benefit of this key contribution.

Along the approach of truncations, reference [53] proposed the **SnAp-n** algorithm that allows the user to customize the amount of truncation by deciding on $n$. SnAP-n stores $\frac{\mathrm{d}\,s_{j,t}}{\mathrm{d}\,w_{pq}}$ (seen in Eq. S8) only for $j$ such that parameter $w_{pq}$ influences the activity of unit $j$ within $n$ time steps. SnAp-1 is closely related to e-prop/RFLO (assuming no autapses). However, starting at $n = 2$ (SnAp-2), $\frac{\mathrm{d}\,s_{j,t}}{\mathrm{d}\,w_{pq}}$ will be stored for every $\{j, p, q\}$ such that $w_{pq}$ influences $j$ in two steps. This would require the storage of $kN^3$ traces, where $k$ is a constant that equals the connection density squared. To our knowledge, there is no evidence on how neural circuits can accommodate such $O(kN^3)$ storage. Therefore, SnAp-n ($n \geq 2$) still poses a significant biological plausibility issue while SnAp-1 reduces to e-prop/RFLO in certain circumstances.

Moving from temporal truncation, **KeRNL** [52] approximates long term dependencies by assuming the dependency to be first order low-pass and learn the parameters using node perturbation. However, the algorithm poses significant implementation and biologically plausibility issues: (1) it uses node perturbation to find the meta-parameters (e.g. the first order decay constant), which is not scalable; (2) meta-parameters are updated per step on the same timescale as synaptic weight update. Their idea of approximating long term dependencies by assuming it follows a certain structure rather than truncating it is what partially inspired our rule. Unlike KeRNL, our approximation can lead to successful learning with fixed meta parameters that are likely updated on the evolutionary timescale in biology (Figure S1).

## D   Cost analysis and biological implementation

### D.1   Cost and interpretation for biologically-plausible implementation of ModProp in Eq. 6

Recall for ModProp, the eligibility trace is combined with total modulatory signals detected:

$$\Delta W_{pq} \propto \text{ET}_{pq} \times \text{TD}_p + \text{ET}_{pq} * \sum_{\alpha \in C} \text{LM}_{\alpha\beta}$$

$$\text{ET}_{pq} * \sum_{\alpha \in C} \text{LM}_{\alpha\beta} = \sum_{s=1}^{S} \text{ET}_{pq,t-s} \times \sum_{\alpha \in C} \text{LM}_{\alpha\beta,s}$$

$$\text{LM}_{\alpha\beta,s} = (\text{affinity } W_{\alpha\beta}^s) \times \sum_{j \in \alpha} \underbrace{\text{TD}_j \times (\text{activity } j)}_{\text{modulatory signal } j}. \tag{S27}$$

We see that the eligibility trace (ET) is brought outside of the double summation of local modulatory (LM) signals. A biological interpretation is that secretion of top-down (TD) learning signals can selectively activate a biochemical process at the post-synaptic neuron, which can then act as a temporal filter on the eligibility trace. The number of filter taps for the underlying biochemical process, $S$, determines the number of steps for credit information propagation.

Here is the computational cost breakdown for Remark 3.2:

- $a_{j,t} = \text{TD}_j \times (\text{activity } j)$ for all $j = 1, .., N$ has $O(N)$ operations.

- $\sum_{j \in \alpha} a_{j,t}$ for $j = 1, ..., N_\alpha$ has $O(N_\alpha)$ operations (assuming $a_{j,t}$ already available, from the last step), where $N_\alpha$ is the number of cells in type $\alpha$.

- $LM_{\beta,s} := \sum_{\alpha \in C} W_{\alpha\beta}^s (\sum_{j \in \alpha} a_{j,t})$ for all $\alpha, \beta = 1, ...C$ and $s = 1, ..., S$ has $O(SC^2)$ operations. Note, this step is the modulatory communication step, where type-specific-approximation of weights can reduce the cost.

- $\sum_{s=1}^{S} \text{ET}_{pq,t-s} \times \text{LM}_{\beta,s}$ for all $p, q = 1, ..., N$ has $N^2$ element-wise multiplications per $s = 1, ..., S$, leading to a total of $O(SN^2)$. Since $\beta$ can be determined from $p$, there is no need to loop over $\beta$ in this step.

Since the cost of the last item dominates, **the computational cost scales as** $O(SN^2)$. For storage cost, ModProp stores $e_{pq,t-S}, \ldots e_{pq,t}$ for every $(pq)$, leading to **a storage cost of** $O(SN^2)$.

### D.2   Cost for biologically-implausible implementation of ModProp

We prove Proposition 1 next, where we discussed a potentially biologically-implausible in silico implementation with lower computational and storage costs than the biologically-plausible version above.

*Proof.* Let us first introduce the following notations:

- $N_\alpha$ denotes the number of cells in type $\alpha$

- $[(W^s)_{\alpha\beta}] \in \mathbb{R}^{N_\alpha \times N_\beta}$ is a matrix repeating the value of scalar $(W^s)_{\alpha\beta}$

- Thus, $[(W^1)_{\alpha\gamma}][(W^s)_{\gamma\beta}] = [N_\gamma (W^1)_{\alpha\gamma}(W^s)_{\gamma\beta}]$

- $G^t_{\alpha,pq} := \sum_{s=1}^{t} (W^s)_{\alpha\beta} \, \mu^{s-1} \, e_{pq,t-s}$

By properties of block matrix product:

$$[(W^{s+1})_{\alpha\beta}] = \sum_\gamma [(W^1)_{\alpha\gamma}][(W^s)_{\gamma\beta}] = [\sum_\gamma N_\gamma (W^s)_{\alpha\gamma}(W^s)_{\gamma\beta}]$$

$$\rightarrow (W^{s+1})_{\gamma\beta} = \sum_\gamma N_\gamma (W^1)_{\alpha\gamma}(W^s)_{\gamma\beta}. \tag{S28}$$

Now, let's find a recursive expression to calculate $Z_{\alpha,pq}^{t+1}$ online:

$$
\begin{aligned}
G_{\alpha,pq}^{t+1} &= \sum_{s=1}^{t+1} (W^s)_{\alpha\beta}\, \mu^{s-1}\, e_{pq,t+1-s} \\
&= \sum_{s=0}^{t} (W^{s+1})_{\alpha\beta}\, \mu^s\, e_{pq,t-s} \\
&= \sum_{s=1}^{t} (W^{s+1})_{\alpha\beta}\, \mu^s\, e_{pq,t-s} + (W^1)_{\alpha\beta} e_{pq,t} \\
&= \sum_{s=1}^{t} (\sum_\gamma N_\gamma (W^1)_{\alpha\gamma}(W^s)_{\gamma\beta})\, \mu^s\, e_{pq,t-s} + (W^1)_{\alpha\beta} e_{pq,t} \\
&= \sum_\gamma N_\gamma (W^1)_{\alpha\gamma} \mu \sum_{s=1}^{t} (W^s)_{\gamma\beta}\, \mu^{s-1}\, e_{pq,t-s} + (W^1)_{\alpha\beta} e_{pq,t} \\
&= \sum_\gamma N_\gamma (W^1)_{\alpha\gamma} \mu G_{\gamma,pq}^{t} + (W^1)_{\alpha\beta} e_{pq,t} \tag{S29}
\end{aligned}
$$

And the overall update is:

$$\Delta W_{pq}|_{ModProp} = \frac{\partial E}{\partial z_{p,t}} e_{pq,t} + \sum_\alpha G_{\alpha,pq}^{t} \sum_{j\in\alpha} \frac{\partial E}{\partial z_{j,t}} h_{j,t} \tag{S30}$$

The second term dominates the cost, for which we need to store $G_{\alpha,pq}^{t}$ for every $\alpha, p, q$. This amounts to $O(CN^2)$ storage cost. To update and attain $G_{\alpha,pq}^{t+1}$, we need $O(C)$ summations and multplications per $G_{\alpha,pq}^{t+1}$, which amounts to $O(C^2 N^2)$ computational cost. The final step of combining $G$ and $\sum_{j\in\alpha} \frac{\partial E}{\partial z_{j,t}} h_{j,t}$, requires $O(CN^2)$ computational cost, which does not dominate the cost.

We note that the specific implementation outlined in the proof of Proposition 1 can significantly reduce the implementation cost, but is likely biologically-implausible, because each synaptic weight update requires the knowledge of all modulatory weights in the network (Appendix D). Moreover, it reduces the cost compared to RTRL ($O(N^3)$ storage and $O(N^4)$ computational complexity) as well as SnAP-2 ($O(d^2 N^3)$ storage and $O(d^2 N^4)$ computational complexity for connection density $d$) [53] significantly if only a few cell types are used. In this work, we used only two cell types ($C = 2$) that map onto the two main cell classes: excitatory and inhibitory. However, ModProp is more expensive (by a constant factor) than e-prop, RFLO and MDGL, which all have $O(N^2)$ storage and $O(N^2)$ computational complexity. However, as mentioned, the performance of these rules are limited due to their severe temporal truncation.

$\square$

## E  Unreasonable effectiveness of synapse-type-specific modulatory backpropagation (through time) weights

We provide the proof for Theorem 1 below:

*Proof.* We first show that $\mathbb{E}[(\epsilon_s)_{ij}] = 0$ for all $s \geq 1$. Note $(W^s)_{\alpha\beta} = \sum_\gamma N_\gamma (W^{s-1})_{\alpha\gamma} W_{\gamma\beta}$ and $(W^s)_{ij} = \sum_k (W^{s-1})_{ik} W_{kj}$ can be calculated recursively.

The base case $s = 1$ is already given in the condition. Suppose $\mathbb{E}[(\epsilon_s)_{ij}] = 0$, for $s + 1$:

$$\mathbb{E}[(\epsilon_{s+1})_{ij}] = \mathbb{E}[(W^{s+1})_{ij} - (W^{s+1})_{\alpha\beta}]$$

$$= \mathbb{E}\left[\sum_k (W^s)_{ik} W_{kj}\right] - \sum_\gamma N_\gamma (W^s)_{\alpha\gamma} W_{\gamma\beta}$$

$$= \mathbb{E}\left[\sum_\gamma \sum_{k \in \gamma} ((W^s)_{\alpha\gamma} + (\epsilon_s)_{ik})(W_{\gamma\beta} + \epsilon_{kj})\right] - \sum_\gamma N_\gamma (W^s)_{\alpha\gamma} W_{\gamma\beta}$$

$$= \sum_\gamma N_\gamma (W^s)_{\alpha\gamma} W_{\gamma\beta} + \sum_\gamma \sum_{k \in \gamma} \mathbb{E}[(\epsilon_s)_{ik}] W_{\gamma\beta} + \sum_\gamma \sum_{k \in \gamma} (W^s)_{\alpha\gamma} \mathbb{E}[\epsilon_{kj}]$$

$$+ \sum_\gamma \sum_{k \in \gamma} \mathbb{E}[(\epsilon_s)_{ik} \epsilon_{kj}] - \sum_\gamma N_\gamma (W^s)_{\alpha\gamma} W_{\gamma\beta}$$

$$= \sum_\gamma \sum_{k \in \gamma} \mathbb{E}[(\epsilon_s)_{ik}] W_{\gamma\beta} + \sum_\gamma \sum_{k \in \gamma} (W^s)_{\alpha\gamma} \mathbb{E}[\epsilon_{kj}] + \sum_\gamma \sum_{k \in \gamma} \mathbb{E}[(\epsilon_s)_{ik}] \mathbb{E}[\epsilon_{kj}]$$

$$= 0. \tag{S31}$$

We now prove the Theorem statement for the scalar output case. The extension to multiple output signals follows identically. Consider the loss decrement after one update, under the (locally) first order loss assumption:

$$\mathbb{E}\left[\Delta E|_{pq,t}\right] = -\eta \mathbb{E}\left[\widehat{\frac{dE}{dW_{pq}}} \frac{dE}{dW_{pq}}\right]$$

$$= -\eta \mathbb{E}\left[(y_t - y_t^*)^2 \left[W_p^{\mathrm{OUT}} e_{pq,t} + \sum_{s,\alpha} \sum_{j \in \alpha} W_j^{\mathrm{OUT}} (W^s)_{\alpha\beta} e_{pq,t-s}\right]\right.$$

$$\left. \times \left[W_p^{\mathrm{OUT}} e_{pq,t} + \sum_{u,\alpha'} \sum_{j' \in \alpha'} W_{j'}^{\mathrm{OUT}} [(W^u)_{\alpha'\beta} + (\epsilon_u)_{j'p}] e_{pq,t-u}\right]\right]$$

$$= -\mathbb{E}[\Gamma_{pq}^2] - \eta \mathbb{E}\left[(y_t - y_t^*)^2 \left[W_p^{\mathrm{OUT}} e_{pq,t} + \sum_{s,\alpha} \sum_{j \in \alpha} W_j^{\mathrm{OUT}} (W^s)_{\alpha\beta} e_{pq,t-s}\right]\right.$$

$$\left. \times \left[\sum_{u,\alpha'} \sum_{j' \in \alpha'} W_{j'}^{\mathrm{OUT}} (\epsilon_u)_{j'p} e_{pq,t-u}\right]\right]$$

$$= -\mathbb{E}[\Gamma_{pq}^2] - \eta \sum_{u,\alpha'} \sum_{j' \in \alpha'} W_{j'}^{\mathrm{OUT}} W_p^{\mathrm{OUT}} \mathbb{E}\left[(\epsilon_u)_{j'p}(y_t - y_t^*)^2 e_{pq,t} e_{pq,t-u}\right]$$

$$- \eta \sum_{s,u,\alpha,\alpha'} \sum_{j \in \alpha, j' \in \alpha'} (W^s)_{\alpha\beta} W_j^{\mathrm{OUT}} W_{j'}^{\mathrm{OUT}} \mathbb{E}\left[(\epsilon_u)_{j'p}(y_t - y_t^*)^2 e_{pq,t-s} e_{pq,t-u}\right]$$

$$\overset{(a)}{=} -\mathbb{E}[\Gamma_{pq}^2] - \eta \sum_{u,\alpha'} \sum_{j' \in \alpha'} W_{j'}^{\mathrm{OUT}} W_p^{\mathrm{OUT}} \mathbb{E}\left[(\epsilon_u)_{j'p}\right] \mathbb{E}\left[(y_t - y_t^*)^2 e_{pq,t} e_{pq,t-u}\right]$$

$$- \eta \sum_{s,u,\alpha,\alpha'} \sum_{j \in \alpha, j' \in \alpha'} (W^s)_{\alpha\beta} W_j^{\mathrm{OUT}} W_{j'}^{\mathrm{OUT}} \mathbb{E}\left[(\epsilon_u)_{j'p}\right] \mathbb{E}\left[(y_t - y_t^*)^2 e_{pq,t-s} e_{pq,t-u}\right]$$

$$\overset{(b)}{=} -\mathbb{E}[\Gamma_{pq}^2] \leq 0, \tag{S32}$$

where $(a)$ follows from the uncorrelatedness condition and $(b)$ follows from the result of (S31). Here, we defined $\Gamma_{pq} := \eta(y_t - y_t^*) \left[ W_p^{\text{OUT}} e_{pq,t} + \sum_s \sum_\alpha \sum_{j \in \alpha} W_j^{\text{OUT}} (W^s)_{\alpha\beta} e_{pq,t-s} \right]$. Then,

$$\mathbb{E}[\Delta E|_t] = -\eta \mathbb{E}\left[ \widehat{\nabla E}^T \nabla E \right] = -\eta \sum_{p,q} \mathbb{E}\left[ \widehat{\frac{dE}{dW_{pq}}} \frac{dE}{dW_{pq}} \right] \le 0. \tag{S33}$$

Moreover, if gradient descent is possible for a network $\hat{\mathcal{N}}$ with weight $W_{ij} = W_{\alpha\beta}, \forall i \in \alpha, j \in \beta$, then $\mathbb{E}[\sum_{p,q} \Gamma_{pq}] < 0$ by definition and $\mathbb{E}[\Delta E|_t] < 0$.

$\square$

We note that with the linear RNN assumption in Theorem 1, **Approximation 1** becomes exact when $\mu = 1$ because the activation derivative is a constant 1 for linear networks. Thus, the proof only only examines the effect of **Approximation 2** (type-specific feedback weight approximation). Also, the Theorem assumes uncorrelatedness for residual weights $\epsilon$, which may not be the case for networks that are not Erdős–Rényi [64]. Despite that, ModProp still leads to performance improvement over existing rules for the tasks examined. Nevertheless, it is important to investigate ModProp across a broad range of tasks in the future.

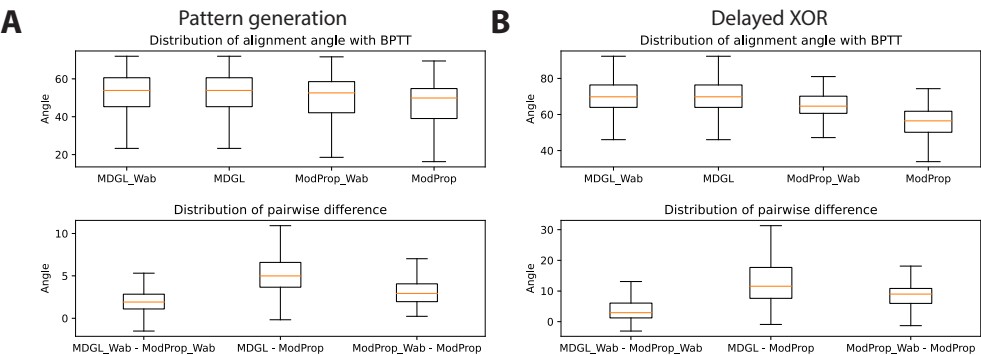

Figure S5: **Alignment angle comparison shows that gradients approximated by ModProp (with or without type-specific modulatory weights) are more similar (than MDGL) to the exact gradients**. We quantify the similarity between approximated and exact gradients via the alignment angle, which describes the similarity in the direction of the two update vectors (Appendix F) for various tasks. In all top-panels, the alignment angles between approximate rules and BPTT are less than $90°$, which indicate that the approximated gradients are aligned with the exact gradient, despite the high-dimensionality of the update vectors. All bottom panel plots show that ModProp variants achieve smaller alignment angles (hence better alignment) with BPTT than MDGL does. To ensure a fair comparison, we examine the statistics of pairwise differences, so that the point on the loss landscape — where the comparison is done — is matched. This is achieved by training the network using BPTT across seven different runs and sampling the approximated gradient once every 50 training iterations. Alignment analysis illustrated here is for recurrent weight gradients, and similar trends are observed for the input weights as well.

# F   Simulation details

All weight updates were implemented using Adam with default parameters [65]. All Adam learning rates are optimized by picking the best one within $\{5e-5, 1e-4, 2e-4, 5e-4, 1e-3, 2e-3, 5e-3\}$ for every learning rule and task. For ModProp, the best value of hyperparameter $\mu$ (Eq. 3) was picked within $\{0.2, 0.25, 0.3, 0.35, 0.4, 0.45, 0.5\}$. For every learning rule and task, we removed the worst performing run quantified by area under the learning curve. We note that while input, recurrent and output weights are all being trained, the nonlocality issue (Eq. S8) only applies to training input and recurrent weights. Thus, all approaches update output weights using backpropagation, and approximations apply to training input and recurrent weights. As stated, we repeated runs with different random initialization to quantify uncertainty and weights were initialized similarly as in [66].

We used alignment angle to quantify the similarity between two vectors. The alignment angle $\theta$ between two vectors, a and b, was computed by $\theta = acos(\|a^T b\|/\|a\|\|b\|)$. The alignment between two 2D matrices was computed by flattening the matrices into vectors.

For the pattern generation task, our network consisted of 400 neurons described in Eq. S1. All neurons had a membrane time constant of $\tau_m = 30$ms. Input to this network was provided by 50 units each producing a different random Gaussian input. The fixed target signal had a duration of 2000ms and given by the sum of five sinusoids, with fixed frequencies of 0.5Hz, 1Hz, 2Hz, 3Hz and 4Hz. For learning, we used mean squared loss function and for visualization, we used normalized mean squared error NMSE $= \frac{\sum_{k,t}(y^*_{k,t}-y_{k,t})^2}{\sum_{k,t}(y^*_{k,t})^2}$ for zero-mean target output $y^*_{k,t}$. For the delayed XOR task, our implementation of the task involved the presentation of two sequential cues, each lasting 100ms and separated by a 700ms delay. There was only one input unit involved and two cue alternatives were presented by setting the input unit to 1 or 0, and the unit was set to to 0 during the delay period. In addition, a Gaussian noise with $\sigma = 0.01$ was added to the input. The network was trained to output 1 (resp. 0) at the last time step when the two cues have matching (resp. non-matching) values. Our network consisted of 120 neurons. All neurons had a membrane time constant of $\tau_m = 100$ms. For learning, we used cross-entropy loss function and the target corresponding to the correct output was given at the end of the trial. A batch size of 32 was used and the gradients were accumulated during those trials additively.

For the copy task, we presented a input sequence of seven binary cues (taking on the value of 0 or 1) on one set of runs and five cues on another. Each cue lasts 100ms to mimic duration of a quick cue flash in biological setting. After the full sequence presentation, the network is tasked to output the same sequence (same value and duration) without further instruction. Our network consisted of 120 neurons. All neurons had a membrane time constant of $\tau_m = 100$ms. For learning, we used cross-entropy loss function and the target corresponding to the correct output was given at the end of the trial. We used full batch training: a batch size of 8 (resp. 128) was used for the three (resp. seven) cue sequence runs due to 8 (resp. 128) possible permutations.

For the pixel-by-pixel MNIST task [58], our network consisted of 200 neurons. All neurons had a membrane time constant of $\tau_m = 20$ms. Input to this network was provided by a single unit that represented the grey-scaled value of a single pixel, with a total of 784 steps and the network prediction was made at the last step. For learning, we used the cross-entropy loss function and the target corresponding to the correct output was given at the end of the trial. A batch size of 256 was used and the gradients were accumulated during those trials additively.

We used TensorFlow [67] version 1.14 and based it on top of [66]. [1] We performed simulations on a computer server with x2 6-core Intel Xeon E5-2640, 2.5GHz, 32 GB RAM. Regardless of the learning rule, our implementation takes approximately one hour to complete one run of pattern generation or delayed XOR task training (for Figures 3 and 4) on the server.

---

[1]Our code is available: `https://github.com/Helena-Yuhan-Liu/ModProp`.