# OpenReview forum: "Biologically-plausible backpropagation through arbitrary timespans via local neuromodulators"
_NeurIPS.cc/2022/Conference — NeurIPS 2022 Accept_

### Official Review · Reviewer_GvaQ · 2022-07-06

**Rating:** 5
**Confidence:** 4
**Soundness:** 2 fair
**Presentation:** 2 fair
**Contribution:** 2 fair

**Summary:**

### UPDATE 08/08/2022: Contribution score increased.


### UPDATE 08/05/2022: Score increased slightly after reading the response and updated version of the submission.


This paper proposes a biologically plausible implementation of training recurrent neural networks as a biologically plausible alternative to back-propagation through time and already existing less plausible algorithms. This works lays out the approximation necessary for making the model believable as to what type of cell could be implementing it. The paper also proposes some experiments to valide the model.

**Questions:**

What predictions can be made based on your model of the modulating candidates?

How do you propose to present the main justification of the important assumptions that make the model plausible?



**Limitations:**

They have addressed some of the limitations.

**Strengths And Weaknesses:**

The paper is well-written, and results and theorems are nicely explained using informal explanations and leaving the full proof to the appendix. The paper is somewhat novel and addresses an important question, which is often less addressed than that of backprop in non-recurrent networks.

Unfortunately, the "framework" proposed seems to be more of a patch of approximations, each solving various problems rather than a mathematically grounded approach. The paper offers a long introduction of what are the possible candidates for their algorithms to justify this work, but it is not validated, nor any predictions are made that could make us believe that it is at all plausible.

The arguments for each approximation are mostly left in the appendix, which is the core of the paper. I would have wanted to be shown in the paper why they are possible and not have to trust the authors and move on. I observe mainly the discrepancy between the claims of the paper and what is actually shown in the paper.

---

> ### Author Response · Authors · 2022-08-02
> **Response to reviewer GvaQ (2/2)**
>
> **“Any predictions made that could make us believe it is at all plausible… How do you propose to present the main justification of the important assumptions that make the model plausible?”**
>
> For justifying the assumptions that make the model biologically plausible (i.e. not violating any known biological constraints), we would like to emphasize that biological plausibility is a guiding principle in developing our model: all of our assumptions are based either on firmly established constraints of neurobiology such as the locality of synaptic transmission, causality, and Dale’s Law, or on emerging evidence from large-scale datasets, such as the cell-type-specificity of certain neuromodulators [Smith et al., 2019] or the hierarchical organization of cell types [Tasic et al., 2018]. To address the reviewer’s concern, we edited the manuscript to emphasize these neurobiological constraints and evidence.
>
> **What predictions can be made based on your model of the modulating candidates? **
>
> We thank the reviewer for the excellent and important question.
>
> Figure 1 caption states that *“... predicts that the modulatory signal each neuron receives can represent a filtered credit signal regarding how its past firing (arbitrary steps back) contributes to the task outcome.”* Our empirical experiments (i.e., learning curves, alignment angle experiment) and theoretical development provide in-silico evidence for this prediction to be validated in future neurobiological experiments. A further prediction of our model is that the neuromodulation acts on its recipients (to signal credit information) in a synapse-type-specific manner. This is due to $\alpha,\beta$ dependence of F in Eq. 6.
>
> We would also like to suggest the outline of an experiment that can test some of the model’s predictions on a family of signaling molecules (e.g., neuropeptides): physiology of multiple individual cells can be monitored in modern neurobiology experiments. Blocking the peptidergic receptors of the neurons that are involved with learning a task and comparing the performance to that without blocking can provide a strong test for the role of peptidergic communication. By changing the task parameters, one could also estimate the temporal extent of credit assignment enabled by peptidergic signaling. In response to the reviewer’s great question, we have now added a discussion on experimental prediction also to the first paragraph in Discussion.
>
> We would also like to remark that equations derived in our work offer potentially testable predictions, since we assign putative identities and mechanisms to the terms that appear in our equations. (e.g., cell-type-specific local neuromodulator, global neuromodulator, activity of cell of interest, activity of neighboring cell, filtering at the postsynaptic site).
>
> **References:**
>
> [Smith et al]  "Single-cell transcriptomic evidence for dense intracortical neuropeptide networks," elife, 2019.
>
> [Tasic et al., 2018]  "Shared and distinct transcriptomic cell types across neocortical areas," Nature, 2018.

---

> > ### Comment · Reviewer_GvaQ · 2022-08-05
> > **Response to the authors.**
> >
> > Thank you for the thorough response. With the new organization of the paper, I can better appreciate the context and justification for the various approximations. I am still not entirely comfortable with the overall writing and organization of the article. I still believe that the work represents a succession of not as significant contributions as the authors claim that, added together, should be of substantial novelty.
> >
> > I am nonetheless happy to increase the score a little.

---

> > > ### Author Response · Authors · 2022-08-06
> > > **Thank you**
> > >
> > > We are very grateful to this reviewer for careful reading of our response and taking that into consideration to revise their score.
> > >
> > > We could not be sure whether the subscores are already revised so if this reviewer does not mind, we would like to bring the 'contribution' subscore to this reviewer's attention, which currently sits at "1-poor".

---

> > > > ### Comment · Reviewer_GvaQ · 2022-08-08
> > > > **Response about contribution score**
> > > >
> > > > Thank you. I have updated the contribution score to 2 to better reflect my overall score.

---

> > > > > ### Author Response · Authors · 2022-08-09
> > > > > **Thank you**
> > > > >
> > > > > Great. We thank the reviewer for this update.

---

> ### Author Response · Authors · 2022-08-02
> **Response to reviewer GvaQ (1/2)**
>
> We thank the reviewer for their time, and for mentioning that the paper is well written and addresses an important question. Below we address all the attendant issues and concerns raised to the best of our understanding.
>
> **A patch of approximations**:  In carefully reviewing our paper, we definitely appreciate the reviewer’s concern and perspective, and see how the presentation could have led to the impression that the approximations lead the algorithm rather than the other way around.  We have now added a sentence before Approximation 1 (Eq. 3) stating that the mathematics of neuromodulatory broadcast leads to the approximations. We would like to emphasize that the overriding framework proposed here is *communicating the credit information via cell-type-specific neuromodulators and processing it at the receiving cells via pre-determined temporal filtering taps*. The implementation of this framework utilizes mean-field type approximations, whose justification comes from known neurobiology constraints: they are needed so that our model can abide by firmly established neurobiology constraints (e.g., locality, causality, Dale’s Law). We would like to remark that our approach — starting from the exact gradient and then introducing approximations that would lead to a biologically plausible learning rule — is a common practice in the area of biologically plausible learning rule (e.g. see [4-6] in the manuscript).  In particular, a central contribution of this paper, in addressing our goal, is to understand how cell-type-specific neuromodulation could contribute to temporal credit assignment, and to model such mechanism, cell-type-approximations were made due to the cell-type-specificity of neuromodulation [Smith et al., 2019].
>
> Related to the comment below, we have now added explanations offering mechanistic intuition of why the approximation works on top of the full Theorem proof in Appendix. We hope that the improved treatment of the approximations and how they are driven by a mechanistic theory will address the reviewer’s understandable concern.
>
> **“The arguments for each approximation are mostly left in the appendix, which is the core of the paper. I would have wanted to be shown in the paper why they are possible and not have to trust the authors and move on. ”**:
>
> We understand the need to more explicitly give arguments in the main text. Unfortunately, we had to defer proofs of the formal statements to the appendix due to space limitations, which is common practice in NeurIPS. The reader does not have to trust the authors and can check the proofs and other supplementary material in the appendix. To address the reviewer’s concern, we have now added explanations offering mechanistic intuition for these approximations in the main text (Section 3.2).
>
> **“I observe mainly the discrepancy between the claims of the paper and what is actually shown in the paper.”**:
>
> We respectfully disagree. If the reviewer could explicitly state the discrepancies between claims vs what is shown, we will be happy to address those issues. We would also like to point out that in response to other reviewer’s comments, we have now expanded the delayed XOR experiment to study the effect of the delay length and included a new task (“copy task”) to demonstrate the superiority of ModProp over existing biological plausible learning rules (in Appendix B). We hope that this expanded set of experimental evidence together with the newly added explanations of the theoretical results will address the reviewer’s concern.
>
> **Please continue to the comment below for part (2/2) of our response.**

---

### Official Review · Reviewer_rAKi · 2022-07-08

**Rating:** 7
**Confidence:** 2
**Soundness:** 3 good
**Presentation:** 2 fair
**Contribution:** 3 good

**Summary:**

The authors propose a method of temporal credit assignment method for training recurrent neural networks, which is an approximation of real time recurrent learning. Their method can be implemented in a biologically plausible neural network with neuromodulators.

**Questions:**

Numerics:

- Why don't you compare your algorithm with RTRL since it's an approximation of RTRL? Wouldn't that help isolate if the performance gap with BPTT is due to the approximations or because RTRL isn't as good as BPTT?

Relation to biology:

- I think it's worth further discussing the relationship of this work to [6]. Both works use cell-type specific neuromodulation, but the similarities and differences of the biological interpretations are not clearly stated.

Derivation: I think the clarify and precision of the technical arguments can be improved, especially for a novice like me who is not familiar with RTRL.

- Should Eq. (1) and Eq. (10) be the same?
- Line 129: What are the mathematical definitions of $\partial$ and $\text{d}$? One involves the chain rule and the other doesn't?
- Line 134: There are a number of forward references to equations in the appendix, which makes the paper difficult to read on its own.
- Eq. (3): Should "$\text{d}$" be "$\partial$"?
- Is $z=\text{ReLU}(s)$? It seems like that needs to be true for Eq. 3 to hold. Why not say so in the main text?
- Line 143: Is it reasonable to assume that neural activities and weights are uncorrelated? Do you have experimental evidence or a reference? My naive impression would be that this is *not* true.
- Line 149: Based on Eq. (1), should $e_{pq,t}$ be $\frac{\text{d}s_{p,t}}{\text{d}W_{pq}}$? And wouldn't that be $\frac{\partial s_{p,t}}{\partial W_{pq}}$ in this case?
- Eq. (11): Should $\frac{\text{d} z_{j,t}}{\text{d} s_{j,t}}$ be $\frac{\partial z_{j,t}}{\partial s_{j,t}}$? Is $h_{j,t}$ simply the derivative of the ReLU function evaluated at $s_{j,t}$?
- Line 568: A more detailed justification for approximation (a) would be greatly appreciated. I see why the approximation is exact in the linear setting. What does it mean for the $W$ and $h$ chains to be uncorrelated? Is that a reasonable assumption in the ReLU setting?

**Limitations:**

The authors make a number of approximations when deriving their algorithm and I am interested to better understand to what extend these approximations affect the performance of their algorithm. In the discussion section, the authors suggest investigating their algorithm in situations where the assumptions are violated.

**Strengths And Weaknesses:**

Strengths:

The authors tackle and challenging and important problem in theoretical neuroscience: biologically plausible training of RNNs (though an area I'm not very familiar with). I find their general approach quite interesting and it appears to suggest (potentially testable) predictions about the role of neuromodulators. Overall, I think it is a useful contribution to the field.

Weakness:

The derivation of their algorithm is highly technical and requires a lot of notation. I found it challenging to follow their arguments. First, the main text includes a number of forward references to the appendices, and then the equations in the main text and appendices are not aligned. Sometimes this appears due to some approximation that is made, but other times it's not clear to me. I think overall their technical arguments could be presented with more clarity. I also think that the authors could devote more space to explaining the approximations that they make, and when these approximations are valid or not.

---

> ### Author Response · Authors · 2022-08-02
> **Response to reviewer rAKi (2/2)**
>
> **Comparison with RTRL instead of BPTT**: We agree that this point needed further clarification and appreciate the reviewer’s concern.  RTRL and BPTT both compute the exact gradient. They differ only in how they parse this computation. Therefore, they should be identical in terms of performance. The reason we chose BPTT here is that RTRL is prohibitively expensive to compute (O(N^3) memory and O(N^4) computation costs). Thus, our choice merely reflects a cost-cutting measure without affecting the results. To address the reviewer’s understandable concern, we have now emphasized this point in the manuscript.
>
> **Relationship to Ref. [6]**: We would like to emphasize that while Ref. [6] introduced the notion of learning with cell-type-specific neuromodulation, their algorithm only addresses the contributions to the task error of neurons that are at most 2 synapses away. Our manuscript shows a way of removing this significant limitation and enables the communication and calculation of credit from neurons that can be arbitrarily many synapses away. It also experimentally demonstrates the benefit of this key contribution. To address the reviewer’s concern, we have carefully revised  the manuscript to pinpoint the mechanistic differences between our work and Ref. [6]. (e.g., Ref. [6] can be considered as a special case of our work, where the filter length is constrained to a single tap.) We have also added Appendix C (due to space limit) to expand the discussion on Ref. [6] and other cited works such as KeRNL and SnAP, and alerted the reader to that section in Related Works.
>
> **Line 143, uncorrelated activities and weights**: The reviewer is correct that neuronal activity and synaptic weights are not uncorrelated, strictly speaking. On the other hand, considering that a single neuron may have thousands of synaptic partners, the activity of the neuron or its time derivative is at best weakly correlated to any one synaptic weight. (e.g., the “trial-to-trial variability” in controlled experiments could perhaps be considered as an example: the firing of individual neurons can demonstrate significant variability under identical experimental conditions even though the synaptic weights are supposed to remain essentially the same.) We take advantage of this phenomenon in our model and ignore these weak correlations. To address the reviewer’s concern, we have now revised the manuscript to spell out this reasoning.
>
> **Line 568, uncorrelatedness of W and h chains**: Since W refers to the synaptic weight and h refers to the derivative of the activity, we would like to refer to the uncorrelatedness argument above for activities and weights. The reviewer is correct in pointing that analysis of nonlinear networks (e.g., with ReLU activations) can be hard. (This is a key reason for the ever-increasing analysis of linear networks in the field.) On the other hand, as mentioned above, neuroscience experiments offer empirical evidence: nonlinearity is a hallmark of neuronal computation. Yet, neuronal activity is at best weakly correlated to the strength of any one synapse, due, in part, to the involvement of many synapses.
>
> As for how such assumption of uncorrelatedness as well as stationarity of activity (Approximation 1) affects learning performance, we empirically demonstrated in Figure 3 that little performance degradation due to such approximation is seen for the studied neuroscience-motivated tasks. On the other hand, as discussed in our submission, such approximation restricts the spatiotemporal precision of the credit signal, making ModProp struggle with tasks that require precise input integration, e.g. sequential MNIST. As we argued, sequential MNIST is a task that would also be difficult for the brain to solve.

---

> > ### Comment · Reviewer_rAKi · 2022-08-04
> > **Response to rebuttal (1/2)**
> >
> > Thank you for your response. I am currently checking your revision; however, in the meantime, I have another comment:
> >
> > - The work by Pogodin and Latham [1] seems similar is spirit to your work. They consider a 3-factor learning rule in a deep-network that avoids backprop by using a layer specific modulation of the updates. Sorry I didn't recall this paper in my initial response.
> >
> > [1] Pogodin, Roman, and Peter Latham. "Kernelized information bottleneck leads to biologically plausible 3-factor Hebbian learning in deep networks." Advances in Neural Information Processing Systems 33 (2020): 7296-7307.

---

> > > ### Author Response · Authors · 2022-08-05
> > > **Discussion with rAKi (2/2)**
> > >
> > > **Line 613: In the ReLU setting, why is the total number of different activation chain combinations equal to the product of the number of activations at each time step? Maybe I am missing something obvious.**
> > >
> > > As for number of activation chain combinations, based on the definition of h-chain in the response above, we are essentially choosing a chain of numbers: at each time step, the number we choose correspond to the index of an activated neuron, and the number of indices we can choose from correspond to the number of activated neurons. Thus, the number of activation chain combinations is equal to the product of the number of activations by analogy to the previous paragraph.
> > >
> > > We are sorry this was not clear in our initial submission, and we hope our more precise definition of h-chain in the response above can help with the clarification. We also added a sentence in the updated manuscript to clarify the reason.
> > >
> > > **Should $N_s$ in Eqs (S13) and (S15) be $N^s$?**
> > >
> > > We thank the reviewer for catching these typos and we have fixed it in the updated manuscript.
> > >
> > > **The work by Pogodin and Latham [1] seems similar in spirit to your work. They consider a 3-factor learning rule in a deep-network that avoids backprop by using a layer specific modulation of the updates.**
> > >
> > > We thank the reviewer for bringing up this very interesting work that addresses the important direction of biologically plausible alternatives to backpropagation (in deep feedforward networks). Indeed, this work seems to fit very nicely in our discussion on 3-factor learning and neuromodulation in Related Works, so we have added [1] to our citation in the updated manuscript.
> > >
> > > [1] Pogodin, Roman, and Peter Latham. "Kernelized information bottleneck leads to biologically plausible 3-factor Hebbian learning in deep networks." Advances in Neural Information Processing Systems 33 (2020): 7296-7307.

---

> > > ### Author Response · Authors · 2022-08-05
> > > **Discussion with rAKi (1/2)**
> > >
> > > We are extremely grateful for the reviewer's additional specific suggestions to improve the clarity of the paper. We apologize for anything that is unclear.
> > >
> > > **What are the mathematical definitions of $\partial$ and $\text{d}$? You provide a brief description and a pointer to Ref. [5]. However, the derivation of your algorithm is rather mathematically involved and the derivatives are central to the derivation, so it would be useful to have precise mathematical definitions of these derivatives in the appendix (rather than pointers to another reference).**
> > >
> > > We certainly agree with the reviewer that this point should be clarified more in the paper, as it is central to our derivation. We have added the following explanation to **Notation for Derivatives** in Appendix A.2: *“Without loss of generality, consider a function $f(x,y)$, where $y$ itself may depend on $x$. The partial derivative $\partial$ of $f$ considers $y$ as a constant, and evaluates as $\frac{\partial f(x,y)}{\partial x}$. The total derivative $d$, on the other hand, takes indirect dependencies into account and evaluates as $\frac{d f(x,y)}{d x} = \frac{\partial f (x,y)}{\partial x} + \frac{\partial f (x,y)}{\partial y} \frac{\partial y}{\partial x}$.”* We also added an explanation of how this results in very different computations for $\frac{\partial s_{p,t}}{\partial W_{pq}}$ vs $\frac{d f(x,y)}{d x}$.
> > >
> > > **Thank you for clarifying your justification for assuming uncorrelated weights and activities. Can you be more mathematically precise about how approximation (a) follows from assuming that $W$ and $h$ chains are uncorrelated and the central limit theorem applies. What is the central limit approximation you are making? What does it mean for the "$W$ and $h$ chains to be uncorrelated"?**
> > >
> > > We thank the reviewer for pointing out this omission. We define a $W$-chain (of length $l$) as
> > > \begin{equation}
> > > \prod_{\phi=1}^{l} W_{i_\phi i_{\phi+1}},
> > > \end{equation}
> > >
> > > for any indices $i_1,\ldots,i_{l+1} \in \{1, ..., N\}$. Similarly, we define an $h$-chain (of length $l’$) as
> > > \begin{equation}
> > > \prod_{\theta=1}^{l'} h_{j_\theta, t-\theta},
> > > \end{equation}
> > >
> > > for any indices $j_1,\ldots,j_{l’} \in \{1, ..., N\}$. With these definitions, we call the $W$-chain $W_{i_1,\ldots,i_{s-1}} = W_{j i_1} W_{i_1 i_2} \ldots W_{i_{s-1} p}$ and the $h$-chain $h_{i_1,\ldots,i_{s-1}} = h_{i_1, t-1} \ldots h_{i_{s-1},t-s+1}$ uncorrelated if
> > >
> > > \begin{equation}
> > > \mathbb{E} [W_{i_1,\ldots,i_{s-1}} h_{i_1,\ldots,i_{s-1}}] = \mathbb{E} [W_{i_1,\ldots,i_{s-1}}] \mathbb{E} [h_{i_1,\ldots,i_{s-1}}], \text{where the expectation is over $i_1,\ldots,i_{s-1}$.}
> > > \end{equation}
> > >
> > > Considering $W_{i_1,\ldots,i_{s-1}}$ and $h_{i_1,\ldots,i_{s-1}}$ as random i.i.d. samples indexed by $i_1,\ldots,i_{s-1}$, the central limit theorem states that
> > > \begin{equation}
> > > \sum_{i_1,\ldots,i_{s-1}} W_{i_1,\ldots,i_{s-1}} h_{i_1,\ldots,i_{s-1}} \sim \mathcal{N}(N^S \mathbb{E}[W_{i_1,\ldots,i_{s-1}} h_{i_1,\ldots,i_{s-1}}], N^S \text{Var}(W_{i_1,\ldots,i_{s-1}} h_{i_1,\ldots,i_{s-1}}))
> > > \end{equation}
> > >
> > > as the sum tends to infinity. Here, we simply use the i.i.d. assumption even though stronger version of the Central Limit Theorem need weaker assumptions than i.i.d. When the $W$- and $h$-chains are uncorrelated, we take the mean of this distribution as a point estimate (note, however, the growing variance) to arrive at the following approximation:
> > > \begin{equation}
> > > \sum_{i_1,\ldots,i_{s-1}} W_{i_1,\ldots,i_{s-1}} h_{i_1,\ldots,i_{s-1}} \approx N^S \mathbb{E}W_{i_1,\ldots,i_{s-1}} h_{i_1,\ldots,i_{s-1}} = N^S \mathbb{E}W_{i_1,\ldots,i_{s-1}} \mathbb{E} h_{i_1,\ldots,i_{s-1}}.
> > > \end{equation}
> > >
> > > Since $\mathbb{E}W_{i_1,\ldots,i_{s-1}} = \frac{1}{N^S} (W^s)_{jp}$ when $i_1,\ldots,i_{s-1}$ are distributed uniformly over valid index ranges, we conclude that
> > >
> > > \begin{equation}
> > > \sum_{i_1,\ldots,i_{s-1}} W_{i_1,\ldots,i_{s-1}} h_{i_1,\ldots,i_{s-1}} \approx (W^s)_{jp} \frac{1}{N^S} \sum_{i_1,\ldots,i_{s-1}} h_{i_1,\ldots,i_{s-1}}.
> > > \end{equation}
> > >
> > > We have also added the above explanation beneath equation (S14) in the updated manuscript.

---

> > > > ### Comment · Reviewer_rAKi · 2022-08-07
> > > > **Continued discussion**
> > > >
> > > > Thank you very much for your clarifications, I better understand the derivation; however, I still have some comments/questions.
> > > >
> > > > - Line 625: Why is $\mathbb{E} W_{i_1,\dots,i_{s-1}}=\frac{1}{N^s}(W^s)_{jp}$? (I'm assuming $N^S$ should be $N^s$).
> > > > - (S20): Are you also assuming $\mathbb{E}h_{i_1,\dots,i_{s-1}}=\frac{1}{N^s}\sum_{i_1,\dots,i_{s-1}}h_{i_1,\dots,i_{s-1}}$? If so, why does that hold?
> > > >
> > > > Overall, I think this paper is of interest to the NeurIPS community, so I am raising my score 1 point. On the other hand, I think this paper still has room for improvement in terms of clarify of presentation though I sympathize with the authors since it is difficult to simultaneously be as mathematically careful as possible and make strong connections to physiology, all in 9 pages.

---

> > > > > ### Author Response · Authors · 2022-08-08
> > > > > **Thank you**
> > > > >
> > > > > We very much appreciate the reviewer for recognizing the significance and interest of this work to the NeurIPS community. We would also like to extend our gratitude toward the reviewer for carefully reading our response and take that into consideration for the score revision. Moreover, improving the paper presentation (while meeting the nine-page constraint) is crucial and we really appreciate all this reviewer’s valuable and specific feedback to help us in that direction.
> > > > >
> > > > > **Why is $\mathbb{E} W_{i_1, …, i_{s-1}} = \frac{1}{N^s} (W^s)_{jp}$? (I’m assuming $N^S$ should be $N^s$).**
> > > > >
> > > > > Consider first the $s=1$ case, which is equivalent to the definition of the product of two matrices:  $(W)^1_{jp} = W_{jp} = \sum_{i_1} W_{j i_1} W_{i_1 p}$.
> > > > >
> > > > > In the general case, $(W)^s_{jp} = \sum_{i_1,\ldots,i_{s-1}} W_{j i_1} W_{i_1 i_2} \ldots W_{i_2 i_{s-1}} W_{i_{s-1} p}$. Recognizing the summand in the right hand side is equal to the definition of the $W$-chain $W_{i_1,\ldots,i_{s-1}}$, we have
> > > > > \begin{equation}
> > > > > (W)^s_{jp} = \sum_{i_1,\ldots,i_{s-1}} W_{i_1,\ldots,i_{s-1}},
> > > > > \end{equation}
> > > > > which is $N^{s-1}$ times the empirical estimate of $\mathbb{E} W_{i_1,\ldots,i_{s-1}}$.
> > > > >
> > > > > We also thank the reviewer for bringing up the typo, as this made us realize that the normalizing denominator should be $N^{s-1}$ instead of $N^s$ (the indices go up to $i_{s-1}$. Thus, we have replaced all $N^S$ with $N^{s-1}$ in the manuscript.
> > > > >
> > > > > **(S20): Are you assuming $\mathbb{E} h_{i_1, …, i_{s-1}} = \frac{1}{N^s} \sum_{i_1, …, i_{s-1}} h_{i_1, …, i_{s-1}}$? If so, why does that hold?**
> > > > >
> > > > > Yes, we again replace the expectation with its empirical estimate (again via the Central Limit Theorem):
> > > > >
> > > > > \begin{equation}
> > > > > \mathbb{E}h_{i_1,\ldots i_{s-1}} \approx \frac{1}{N^{s-1}} \sum_{i_1,\ldots,i_{s-1}}h_{i_1,\ldots i_{s-1}}
> > > > > \end{equation}
> > > > >
> > > > > We have now made this point explicit in the updated manuscript.

---

> > ### Comment · Reviewer_rAKi · 2022-08-04
> > **Response to rebuttal (2/2)**
> >
> > I am still having trouble following some steps in the derivation and could use more mathematical hand holding.
> >
> > - What are the mathematical definitions of $\partial$ and $\text{d}$? You provide a brief description and a pointer to Ref. [5]. However, the derivation of your algorithm is rather mathematically involved and the derivatives are central to the derivation, so it would be useful to have precise mathematical definitions of these derivatives in the appendix (rather than pointers to another reference).
> >
> > - Thank you for clarifying your justification for assuming uncorrelated weights and activities. Can you be more mathematically precise about how approximation (a) follows from assuming that $W$ and $h$ chains are uncorrelated and the central limit theorem applies. What is the central limit approximation you are making? What does it mean for the "$W$ and $h$ chains to be uncorrelated"?
> >
> > - Line 613: In the ReLU setting, why is the total number of different activation chain combinations equal to the product of the number of activations at each time step? Maybe I am missing something obvious.
> >
> > - Should $N_s$ in Eqs (S13) and (S15) be $N^s$?

---

> ### Author Response · Authors · 2022-08-02
> **Response to reviewer rAKi (1/2)**
>
> We thank the reviewer for the supportive comments on our method and for mentioning that it suggests potentially testable predictions about the role of neuromodulators. We also thank the reviewer for highlighting the importance of the problem and the challenge it represents in the field.
>
> **Clarity of technical arguments**: We thank the reviewer for multiple specific suggestions to significantly improve the presentation. We have tried to address all of them. Specifically, in addition to the answers to specific questions below, we have now revised the manuscript, where we
>
> - Made equation (1) and (S4) (previously (10)) the same for consistency
>
> - Added a brief explanation for the ∂ vs d notation in the beginning of Appendix A.2 (due to space limit) and alerted the reader to that in the main text after the first appearance of the notation. Briefly, ∂ denotes direct dependency and d accounts for all (direct and indirect) dependencies, following the notation in [5].
>
> - In relation to the point above, fixed the typos in (3), $e_{pq,t}$ and (S5) (formerly (11)): changed d to ∂ as the reviewer correctly pointed out. Also, the reviewer’s understanding is correct that $h_{j,t}$ is simply the derivative of the ReLU function evaluated at $s_{j,t}$.
>
> - Removed the forward equation reference in line 134
>
> - Made another mention of z=ReLU(s) beneath (3). As a side note, we mentioned the use of ReLU activation in Discussion in our initial submission.
>
> **Please continue to the comment below for part (2/2) of our response.**

---

### Official Review · Reviewer_2iDW · 2022-07-11

**Rating:** 7
**Confidence:** 3
**Soundness:** 3 good
**Presentation:** 3 good
**Contribution:** 3 good

**Summary:**

Evaluating the parameter gradients of recurrent neural networks forward in time ([17]) is both costly and not biologically plausible. This paper introduces novel approximations to the synaptic weight gradients of recurrent neural networks, which can be evaluated forward in time, using procedures that are both less costly and more biologically plausible than full RTRL. The authors discuss possible biological implementations based on plasticity modulation and compare the performance of their method to relevant baselines in a number of experiments.

**Questions:**

- How would the matrix power which appears in the weight update be calculated in a biological network? Even when averaging over connections of the same type, it's unclear to me how plausible this operation is.

- Can the authors discuss in more detail how to approach the problem of determining cell types? For example, could it make sense to group by time constants, when there is heterogeneity in time constants? Can the authors provide some heuristics or intuitions on how they expect such choices to affect their rule?

- Can the authors discuss KeRNL and SnAp in more detail?

- As discussed in the strengths & weaknesses section, can the authors extend their experimental evaluation to more clearly demonstrate the superiority over e-prop/RFLO, including the random modulatory version?

**Limitations:**

Apart from the relatively weak experimental evaluation of the proposed approximations, the authors have done a good job in discussing the limitations of the current work.

**Strengths And Weaknesses:**

Finding biologically plausible learning rules which improve upon e-prop/RFLO is a very important research topic. The decomposition of the gradient and approximations proposed in this paper are a sensible approach forward, similar in spirit (but different from) recent algorithms such as SnAp and KeRNL.

My major concern with the current paper is the somewhat weak experimental evaluation. I leave some additional comments for the authors below, on points where I think the paper could be improved:

- The presentation of section 3.1 could be improved. In particular the proposed biological implementation (an essential point of the present work) should be clarified and discussed in greather depth. [Small additional note: there is a problem with equation references; for example, in line 140, Eq. 14 (supplemental material) is referenced instead of Eq. 2.]

- Weakness of experimental part of the paper: the paper would substantially improve if the authors demonstrated the superiority (and the cases where the different approximations fail) of modprop over e-prop in benchmarks that are not as toyish as pattern generation and delayed XOR, but perhaps not as difficult as sequential MNIST. Perhaps the copy task (with varying sequence length) is a good additional synthetic benchmark? In particular, it would be very good to assess how well the update with random modulatory weights performs in more difficult problems. Given that the focus of this work is on introducing a new approximation to the full gradient on RNNs, the somewhat weak experimental study is the major current weakness of the paper.

- Given that there is a section on efficient computer implementations, a small application (or at least a discussion) to LSTM-like models and a comparison to algorithms such as SnAp (which also reduce the bias compared to e-prop/RLFO) would also enrichen the paper and help understand how well the approximations introduced here impact performance, as well as show their potential usefulness in machine learning, beyond their biological plausibility merits.

---

> ### Author Response · Authors · 2022-08-02
> **Response to reviewer 2iDW (2/2)**
>
> **References:**
>
> [Campagnola et al]  "Local connectivity and synaptic dynamics in mouse and human neocortex," Science, 2022.
>
> [Smith et al]  "Single-cell transcriptomic evidence for dense intracortical neuropeptide networks," elife, 2019.
>
> [Bugeon et al] "A transcriptomic axis predicts state modulation of cortical interneurons," Nature, 2022.
>
> [Schneider et al] "Transcriptomic cell type structures in vivo neuronal activity across multiple time scales," bioRxiv, 2022.
>
> [Gouwens et al] "Integrated morphoelectric and transcriptomic classification of cortical GABAergic cells," Cell, 2020.
>
> [Gala et al] "Consistent cross-modal identification of cortical neurons with coupled autoencoders," Nature computational science, 2021.

---

> > ### Comment · Reviewer_2iDW · 2022-08-08
> > **Rebuttal acknowledgement**
> >
> > Thank you for the replies to my questions and for the new experiments.
> > The paper would have improved more drastically if the authors had presented results for non-synthetic (toy) tasks and stronger relevant baselines such as SnAp-2. I'm therefore not yet comfortable in recommending this paper strongly for acceptance, but I have raised my score by one point.

---

> > > ### Author Response · Authors · 2022-08-09
> > > **Thank you**
> > >
> > > We are very grateful to this reviewer for careful reading of our response and taking that into consideration to revise their score. We agree with the reviewer that the paper would have improved more drastically if harder tasks and stronger benchmarks were used. We would also like to make a side note for future readers of OpenReview (not for the purpose of this review) that the main reason why we did not include SnAp-2 in our comparisons was due to its biological plausibility concerns explained in Appendix C (we focused on comparing against learning rules that are biologically plausible). We thank the reviewer again for all their constructive feedback that led to the improvement of this paper.

---

> ### Author Response · Authors · 2022-08-02
> **Response to reviewer 2iDW (1/2)**
>
> We thank the reviewer for supportive comments on the importance of the problem and the potential promise of our approach, and agree with their  highlighting of the low-cost aspect of the proposed ModProp method.
>
> **Somewhat weak experimental evaluation**: We appreciate this point and agree that further experimental validation was called for to strengthen our paper.  In our revision, we have added further experiments that address the reviewer request in two ways:
>
> - We have taken this reviewer’s suggestion and implemented the copy task with different sequence lengths. We have also studied the effect of random modulatory weights, as suggested by the reviewer. We edited the manuscript to include these experiments (Appendix Figure S4 and referenced in the main text). Briefly, ModProp outperforms other biologically plausible alternatives for the copy task at different sequence lengths, despite using random modulatory weights.
>
> - We performed additional experiments with the delayed XOR task by increasing the delay duration, which increases the difficulty of the underlying task. We edited the manuscript to include these experiments (Appendix Figure S3). Briefly, these results suggest that at an increased delay period, ModProp still outperforms other biologically plausible rules for RNNs.
>
> Overall, these two experiments further strengthen our basic conclusions that ModProp can outperform other biologically plausible alternatives across a range of RNN tasks with varying difficulties.
>
> **Presentation of Section 3.1**: We thank the reviewer for pointing this out. In response, in our revision we improved the presentation of this section by: (1) fixing the equation referencing issue the reviewer pointed out; (2) adding a brief explanation for the biological implementation and referencing Appendix D.1 for greater detail; (3) added bolded subheadings for navigation in Section 3.1.
>
> **KeRNL/SnAp discussion**: We thank the reviewer for the great suggestion. We have now added a detailed discussion of these algorithms, pointing out the main differences of our work from these methods in Appendix C. In Related Works in the main text, we have carefully alerted the reader the contents of that Appendix section.
>
> **Efficiency of different algorithms**: We agree with the reviewer’s comment. To address this concern, we have now expanded upon the existing discussion and include a new comparative paragraph on the complexities of the SnAP, e-prop, and RFLO algorithms at the end of Appendix D.
>
> **Implementation of matrix powers**: Online implementation of matrix powers is indeed costly and not biologically plausible. Our theory suggests that this computation can be accurately approximated without such online calculations when neurons within a cell type have similar synaptic connectivity. In this case, the matrix powers of weight averages can be pre-calculated. In biological terms, noting that the entries of the matrix powers are used as the values of different filter taps, these can be genetically encoded as part of the cell type identity and optimized over evolutionary time scales. Thus, while the individual needs to tune the synaptic weights in an experience-dependent manner, the modulatory mechanism (and the corresponding matrix powers) do not need to be updated with a similar frequency (as backed by the superior performance of even the random fixed modulatory weights in our simulations). We finally note that such filtering mechanisms are ubiquitous in both the cell and the synapse. We thank the reviewer for the question and we have now added aspects of the above discussion to the main text (in Section 3) to clarify this point.
>
> **Determining cell types**: We think this is a great question. We had to remove a relevant discussion in the original submission due to space limitations. Multiple studies suggest that cells of the same type demonstrate consistent properties across a wide range of features, including synaptic connectivity, modulatory connectivity, molecular identity, in-vivo activity, morphology, and intrinsic electrophysiology (e.g., synaptic time constants) [Campagnola et al., Smith et al, Bugeon et al., Schneider et al., Gouwens et al., Gala et al.]. Therefore, our understanding is that one would end up with similar groupings of cells, somewhat independent of the particular grouping criteria. Here, we use two cell types with consistent wiring and type of connectivity (i.e., capturing the main excitatory/inhibitory division as well as consistent synaptic and modulatory connectivity) in our relatively simple models. To address the reviewer’s question, we have now incorporated aspects of this discussion into the main text (in Section 3).

---

### Official Review · Reviewer_fMfy · 2022-07-11

**Rating:** 7
**Confidence:** 3
**Soundness:** 3 good
**Presentation:** 2 fair
**Contribution:** 3 good

**Summary:**

In this paper, the authors propose a biologically-plausible temporal credit assignment rule for recurrent neural networks called ModProp. This work is motivated by recent experimental evidence on the presence of local neuromodulatory networks in the brain. Here, they propose that these synapse-type-specific local modulatory signals are received via low-pass filtering of eligibility traces at post-synaptic neurons.
Using the framework of discrete-time rate-based RNNs, they derive the ModProp synaptic weight update rule and describe its properties. They also provide simulation results comparing the performance of ModProp vs. other biologically-plausible learning rules and Backprop through time on three temporal processing tasks. ModProp outperforms other previously proposed learning approaches, indicating that it is a promising candidate for understanding how the biological networks perform temporal credit assignment.



**Questions:**

- Have you tried the delayed XOR task with different delays between the cues? How would the performance of ModProp vary as a function of this delay?


**Limitations:**

The authors adequately address the limitations and the potential long-term societal impact of their work in the discussion section.

**Strengths And Weaknesses:**

This paper addresses the important question of how credit assignment might occur in biological recurrently connected neural networks. Here they explore the potential role of local neuromodulatory signaling mechanisms and propose the ModProp learning rule. The idea of synapse-type-specific local modulatory signaling complementing Hebbian learning and global neuromodulation does seem like a viable theory. Backed by the empirical results presented here, I think ModProp is a promising candidate for biologically plausible learning in recurrent networks and warrants further exploration.

Although ModProp can technically perform temporal credit assignment over arbitrarily long durations, the influence of distant events in time is negligible. This is akin to the problem of vanishing gradients in backpropagation through time, which makes ModProp ill-suited for very long-term credit assignment. The authors do mention this drawback in the discussion section.

I think this paper is quite constrained by the page limit. Despite this constraint, I believe the authors have done a reasonably good job of describing the important concepts involved in the proposed learning rule.
Minor suggestions: the main paper refers to certain equations (e.g. 14) and figures that are in the supplementary material. It might be helpful for the readers to point out that this content is in the methods/supplementary material.

---

> ### Author Response · Authors · 2022-08-02
> **Response to reviewer fMfy**
>
> We thank the reviewer for supportive comments on the importance of the problem and the potential promise of our approach.
>
> **Very long-term credit assignment**: The reviewer is correct in observing that the influence of distant events in time is small. (As the reviewer also mentions, BPTT suffers from a similar problem and BPTT implementations typically put a hard limit on the temporal window size.) This phenomenon may be related to well-known distinctions in biology between short- and long-term memory mechanisms. The circuits implemented in our manuscript are perhaps best seen as counterparts to short-term (working) memory. If these circuits are complemented by a distinct long-term memory architecture (e.g., mimicking the hypothesized roles of brain structures such as the hippocampus and the entorhinal cortex. Also see ref. [59] in the manuscript.), it may become possible to perform efficient credit assignment with ModProp for events distant in time as well.
>
> **Referring to supp. material**: Thank you for this suggestion, which we have implemented as suggested -- we now both explicitly refer to the Appendix and use a separate numbering system for supplemental equations and figures. (e.g., Appendix Eq. S14)
>
> **Delayed XOR task with different delays**: We appreciate this important question -- and as the reviewer requested, we have now performed an additional experiment with a longer time delay in the delayed XOR task. We have edited the manuscript to report these experiments (new Appendix Figure S3 and reference in the main text). Briefly, these new results suggest that at an increased delay period, ModProp still outperforms other biologically plausible rules for RNNs.  At the same time, its learning performance degrades compared to its performance at a shorter delay period. We believe this observation, now included in the paper, strengthens the paper by reinforcing the  reviewer's important point on vanishing gradients as well as the discussion point on short-term vs long-term memory mechanisms which the reviewer also raised.

---

> > ### Comment · Reviewer_fMfy · 2022-08-09
> > **Response to rebuttal**
> >
> > Thank you for the response to my questions and the addition of the suggested experiment.
> > I have raised my overall rating by 1 point.

---

> > > ### Author Response · Authors · 2022-08-09
> > > **Thank you**
> > >
> > > We are very grateful to this reviewer for reading our response and taking that into consideration to revise their score. We thank the reviewer again for all their constructive feedback that led to the improvement of this paper.

---

### Author Response · Authors · 2022-08-02
**General Response**

We thank the reviewers for their time, insightful summaries and contextualization of our work, and constructive feedback toward its improvement. Through changes (also completely described below) made in our uploaded revision, we  believe we managed to address all the concerns raised by the reviewers, and bring their points to light for readers as well. In particular, we have now added two new requested experiments  (Appendix Figures S3 and S4) — one increasing the difficulty of an existing task and the other a completely new experiment (the ‘copy task’) — both of which address specific reviewer concerns and support and extend our original conclusions.   The changes to the manuscript are colored in blue in the present version for easy referencing. Overall, we believe that these changes have significantly improved both the content and presentation of our submission and appreciate the reviewers’ ideas and time in inspiring them.

---

> ### Author Response · Authors · 2022-08-09
> **Removed blue coloring on August 9th**
>
> We have updated the submission to remove all blue coloring of texts on August 9th.

---

### Meta-Review · Area_Chair_sxg2 · 2022-08-24

**Recommendation:** Accept
**Confidence:** Certain

**Metareview:**

Biologically-plausible backpropagation through arbitrary timespans via local neuromodulators

The authors propose a biologically plausible method for temporal credit assignment called ModProp. They apply their framework on rate-based recurrent neural networks (RNNs), and show that it outperforms previous approaches.

All reviewers acknowledge that this work studies an interesting topic in computational neuroscience. The authors present compelling experimental results on synthetic data sets.

Weaknesses:
- Long-term dependencies cannot be tackled by this approach, which is quite common also for related approaches.
- Somewhat weak experimental evaluation. Evaluation on more complex standard data sets would be beneficial.
- The arguments for the used approximations is left in the appendix.

In general, an interesting study with good experimental results. I propose acceptance.

**Award:**

No

---

### Decision · Program_Chairs · 2022-09-14

Accept